EMBO
Molecular Medicine

# Deubiquitination of RIPK3 by OTUB2 potentiates neuronal necroptosis after ischemic stroke

Fuqi Mei [1,2], Deyu Deng[1,2], Zijun Cao[1,2], Liyan Lou[1,2], Kangmin Chen[1,2], Minjie Hu[2], Zhenhu Zhu[2], Jiangyun Shen [2], Jianzhao Zhang[1,2], Jie Liang[3], Jingyong Huang [4], Min Bao [5], Ari Waisman [6] & Xu Wang [1,2✉]

## Abstract

As a common and severe cerebrovascular disease, ischemic stroke casts a significant shadow over global health. Unfortunately, the mechanisms regulating neuronal death in the affected areas remain largely unclear. Here, we found that deletion of the deubiquitinating enzyme Otubain-2 (OTUB2) significantly alleviated ischemia-induced cerebral infarction and neurological deficits, accompanied by a reduction in neuronal loss, glial activation, and neuroinflammation. OTUB2 was predominantly expressed in neurons and its deletion decreased receptor-interacting protein kinase 3 (RIPK3)-mediated neuronal necroptosis. Moreover, OTUB2 increased RIPK3 protein abundance by inhibiting the proteasomal degradation of RIPK3. Mechanistically, OTUB2 removed K48-linked polyubiquitin chains from RIPK3 through its active site C51. Importantly, pharmacological inhibition of OTUB2 alleviated ischemic brain injury in mice and reduced oxygen-glucose deprivation-induced neuronal death in human brain organoids. These results demonstrate that OTUB2 critically regulates ischemic stroke injury by potentiating neuronal necroptosis, suggesting that OTUB2 inhibition may become a potential therapeutic approach for treating ischemic stroke.

Keywords Ischemic Stroke; Necroptosis; Ubiquitination; OTUB2; RIPK3
Subject Categories Neuroscience; Vascular Biology & Angiogenesis

## Introduction

Ischemic stroke is a leading cause of disability and death, killing around 3 million people annually in the world (Fan et al, 2023). After cerebral ischemia, the interrupted supply of oxygen and glucose induces programmed cell death (PCD) in neurons mediated by TNF receptor type 1 (TNFR1) and FAS, causing irreversible loss of neurons (Martin-Villalba et al, 2001). Among PCD mechanisms,

necroptosis, a type of programmed necrosis, is rapidly induced after ischemic stroke (Degterev et al, 2005; Naito et al, 2020). During necroptosis, cells burst and leak pro-inflammatory contents, which then activate brain-resident or -infiltrating immune cells to incite neuroinflammation (Huang et al, 2024; Newton et al, 2024). As such, inhibition of necroptosis by the small-molecule inhibitor Necrostatin-1 (Nec-1) significantly attenuates ischemic brain injury (Degterev et al, 2005).

After ischemic stroke, receptor-interacting protein kinase 1 (RIPK1) is phosphorylated and activated in the affected neurons. Of note, the kinase activity of RIPK1, which is specifically inhibited by Nec-1, is critical for transducing necroptotic signaling (Degterev et al, 2008). Activated RIPK1 subsequently recruits and interacts with RIPK3, leading to the phosphorylation and activation of RIPK3 (He et al, 2009; Yuan and Ofengeim, 2024; Zhang et al, 2009). Thereafter, activated RIPK3 induces the phosphorylation of mixed lineage kinase domain-like pseudokinase (MLKL), which then oligomerizes and translocates to the organelle or plasma membranes, causing the rupture of membranes (Sun et al, 2012; Wang et al, 2014).

In addition to phosphorylation, the RIPK1-RIPK3-MLKL necroptotic signaling is also critically regulated by ubiquitination, another type of post-translational modification. Ubiquitination controls diverse biological activities by regulating the stability, localization, or activity of protein substrates (Liu et al, 2022; Ruan et al, 2022). For example, decoration of RIPK1 by M1- and K63-linked polyubiquitin chains diverts the downstream cell death signaling to pro-survival and pro-inflammatory NF-κB signaling (Ai et al, 2024; Mahoney et al, 2008). Besides, physiological ubiquitination of RIPK3 at K5 supports the formation of RIPK1-RIPK3 complexes and thereby promotes necroptosis (Onizawa et al, 2015). Ubiquitination is a reversible process that is catalyzed by E1, E2, and E3 ubiquitinating enzymes and counter-regulated by deubiquitinating enzymes (DUBs) (Huang et al, 2024; Ruan et al, 2022). Considering the importance of ubiquitination in necroptotic signaling, necroptosis is intimately regulated by DUBs. For instance, the DUB A20 inhibits necroptosis by disrupting RIPK1-RIPK3 interaction through deubiquitinating RIPK3 (Onizawa et al, 2015).

[1]School of Pharmaceutical Sciences, Wenzhou Medical University, 325035 Wenzhou, China. [2]Oujiang Laboratory (Zhejiang Lab for Regenerative Medicine, Vision, and Brain Health), 325000 Wenzhou, China. [3]Department of Rehabilitation, Central Hospital of Jinhua City, 321000 Jinhua, China. [4]Department of Vascular Surgery, The First Affiliated Hospital of Wenzhou Medical University, 325015 Wenzhou, China. [5]Oujiang Laboratory, The First Affiliated Hospital of Wenzhou Medical University, 325035 Wenzhou, China. [6]Institute for Molecular Medicine, Johannes Gutenberg University Mainz, 55131 Mainz, Germany. ✉E-mail: sunrim@163.com

In recent years, DUBs have emerged as key regulators in ischemic stroke injury (Huang et al, 2024). In this study, we found that ischemic cerebral injury in mice was tightly controlled by the DUB Otubain-2 (OTUB2). Similar to A20, OTUB2 is also a DUB belonging to the ovarian tumor protease subfamily. Previous reports have revealed the multifaceted functions of OTUB2 in cancer (Chang et al, 2022; Ren et al, 2024; Yu et al, 2022). In addition, a recent study shows that OTUB2 ameliorates experimental colitis by enhancing the protective NOD2 signaling (Du et al, 2024). Here, we discovered that genetic ablation of OTUB2 significantly ameliorated cerebral injury after ischemic stroke. Functionally, OTUB2 increased neuronal death after ischemic insult by enhancing RIPK3-mediated necroptosis. Mechanistically, OTUB2 inhibited the degradation of RIPK3 through removing K48-linked polyubiquitin chains from RIPK3, thereby enhancing RIPK3 abundance and activation. Furthermore, pharmacological inhibition of OTUB2 attenuated ischemic stroke injury in mice and neuronal death in human brain organoids. Therefore, our results provide a novel regulatory mechanism for necroptosis and identify a potential therapeutic target for ischemic stroke.

## Results

### OTUB2 contributes to ischemic stroke injury in mice

We analyzed the expression of OTUB2 in C57BL/6 mice and found that the brain expressed the highest amount of OTUB2 (Fig. 1A), indicative of a potential role of OTUB2 in the brain. To decipher the pathophysiological function of OTUB2, we generated $Otub2^{-/-}$ mice, in which OTUB2 was efficiently deleted in the brain (Fig. 1B). Ablation of OTUB2 did not affect the growth of mice and adult $Otub2^{-/-}$ mice had normal brain structures (Fig. 1C,D; Appendix Fig. S1A–D). Besides, OTUB2 deficiency did not alter cerebral blood flow, partial pressure of oxygen, and blood pressure in adult mice (Appendix Fig. S1E–I). However, after middle cerebral artery occlusion (MCAO), the infarct volume in $Otub2^{-/-}$ mice was significantly smaller than that of $Otub2^{+/+}$ mice (Fig. 1E,F), indicating that OTUB2 deficiency attenuates ischemic stroke injury. Consistently, MCAO-induced neurological deficits were also significantly ameliorated in $Otub2^{-/-}$ mice as compared with $Otub2^{+/+}$ mice (Fig. 1G–K). Aging is a significant risk factor for stroke. Therefore, we analyzed the impact of OTUB2 on ischemic stroke injury in aged mice and found that OTUB2 deficiency significantly reduced MCAO-induced cerebral infarction and neurological deficits in aged mice (Appendix Fig. S2A–C). Thus, although OTUB2 has a negligible effect on brain function under physiological conditions, it contributes to cerebral injury after ischemic stroke.

### OTUB2 increases neuronal loss and neuroinflammation after ischemic stroke

Neuronal death is a central event in stroke and directly leads to neurological deficits. In the brain, OTUB2 was predominantly expressed in neurons (Fig. 2A), implying that OTUB2 may affect ischemic stroke injury by modulating neurons. Two days after MCAO, the two neuron-associated markers PSD95 and NeuN were strongly reduced in the infarct brain of $Otub2^{+/+}$ mice and the reduction was attenuated in $Otub2^{-/-}$ mice (Fig. 2B–D), indicating

that OTUB2 deficiency ameliorates ischemia-induced neuronal loss. Consistently, in comparison to $Otub2^{+/+}$ mice, $Otub2^{-/-}$ mice had significantly more neurons in the ischemic penumbra, as evidenced by immunofluorescence staining (Fig. 2E,F). In addition, the immunofluorescence staining found that both the number and activation of microglia were markedly reduced in the ischemic penumbra of $Otub2^{-/-}$ mice (Fig. 2G,H). As brain-resident macrophages, microglia actively participate in ischemic stroke injury by establishing neuroinflammation (Borst et al, 2021; Shichita et al, 2023). Lower levels of pro-inflammatory cytokines and chemokines were detected in the ischemic brain of $Otub2^{-/-}$ mice on day 2 after MCAO (Fig. 2I–M), which is in accordance with the reduced number and activation of microglia in the ischemic penumbra of these mice. Therefore, OTUB2 also has the potential to influence ischemic stroke injury by regulating microglia. However, $Otub2^{+/+}$ and $Otub2^{-/-}$ microglia produced comparable levels of pro-inflammatory cytokines and chemokines upon stimulation with lipopolysaccharide (LPS) (Appendix Fig. S3A–E). Based on these results, we postulate that the reduced microglial proliferation and activation is the consequence, rather than the cause, of the mitigated cerebral injury in $Otub2^{-/-}$ mice after MCAO. Considering the high abundance of OTUB2 in neurons, OTUB2 may directly regulate neuronal death in ischemic stroke injury.

### OTUB2 potentiates neuronal necroptosis after ischemia

To investigate the effect of OTUB2 on neuronal death, we analyzed cell death signaling in the ischemic cerebral hemisphere. Four hours after MCAO, the RIPK1-RIPK3-MLKL necroptotic signaling was activated in the ischemic brain but the Caspase 3-mediated apoptotic signaling remained inactive at this early time point (Fig. 3A–D; Appendix Fig. S4A), which is consistent with a previous report that ischemia sequentially induces neuronal necroptosis and apoptosis in a temporally specific mode (Naito et al, 2020). Twenty-four hours after MCAO, apoptosis was actively ongoing in the ischemic cerebral hemisphere, but it was not affected by OTUB2 deficiency (Appendix Fig. S4B–D), indicating that OTUB2 has no effect on apoptosis. To study the impact of OTUB2 on neuronal death in vitro, we ablated OTUB2 in the HT22 hippocampal neuronal cell line, which is widely used for studying neuronal death (Feng et al, 2014; Karuppagounder et al, 2016). OTUB2 deficiency did not change HT22 cell viability upon treatment with tumor necrosis factor-α (TNF-α) + cycloheximide (CHX), which specifically induces apoptosis (Appendix Fig. S4E). Although RIPK1 phosphorylation was not changed, OTUB2 deficiency significantly impaired the phosphorylation of RIPK3 and MLKL (Fig. 3A–D). Moreover, the protein abundance of RIPK3 was significantly reduced in OTUB2-deficient brains (Fig. 3A,F), suggesting that OTUB2 may modulate the necroptotic signaling by targeting RIPK3. In good agreement with the Western blot results, MLKL phosphorylation was significantly reduced in neurons of $Otub2^{-/-}$ mice after MCAO (Fig. 3H,I), showing that OTUB2 deficiency attenuates neuronal necroptosis. Since necroptosis typically triggers inflammation in vivo due to the release of intracellular contents from necroptotic cells, the reduced necroptosis in $Otub2^{-/-}$ mice was accompanied by decreased neuroinflammation (Fig. 3J–L). Due to the mitigated necroptosis in OTUB2-deficient neurons at the early stage (Fig. 3A–I), cerebral

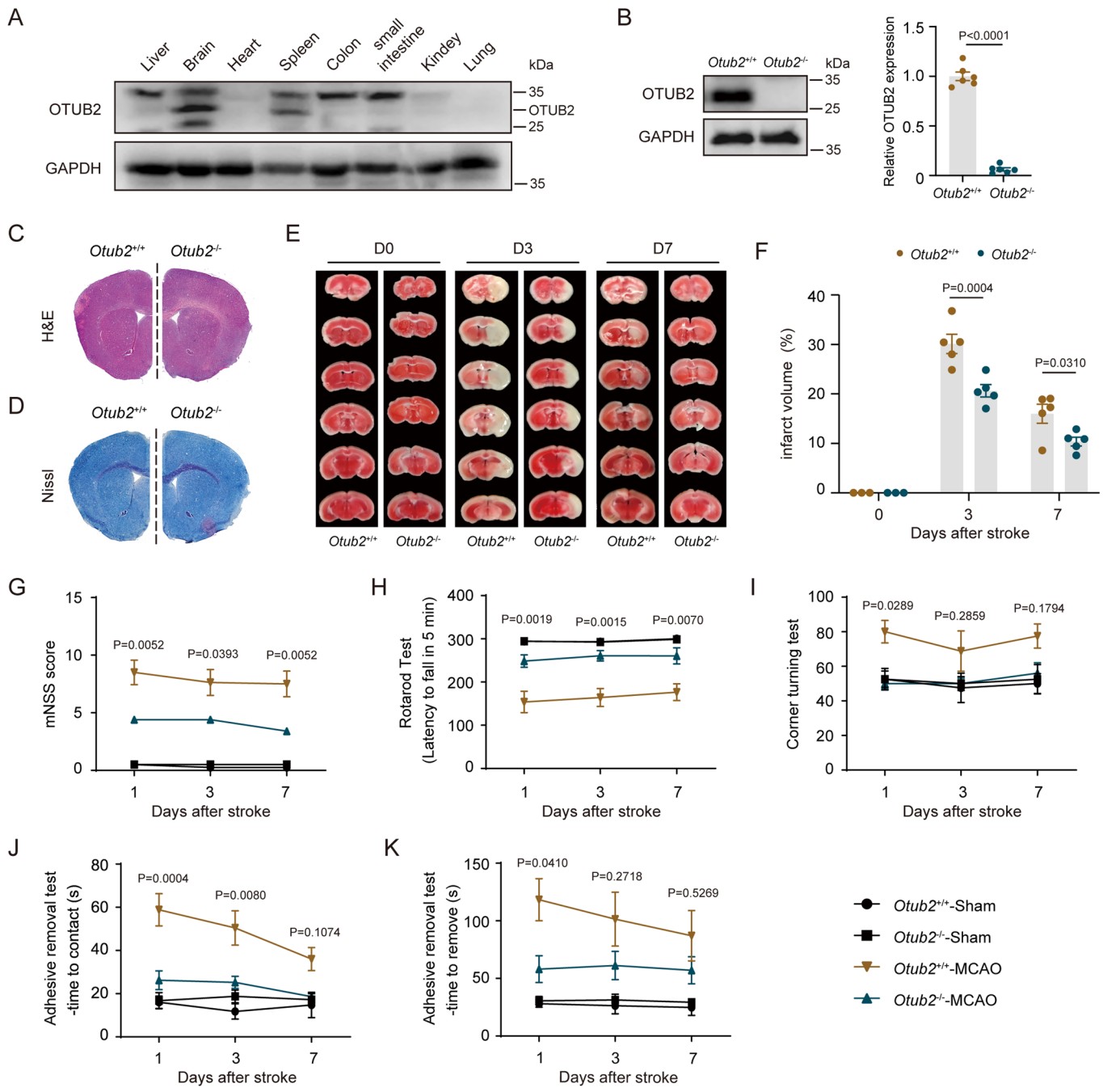

**Figure 1. OTUB2 deficiency attenuates MCAO-induced cerebral injury.**

(A) OTUB2 expression in indicated organs of C57BL/6 mice was determined by Western blot. (B) OTUB2 expression in the brain of $Otub2^{+/+}$ and $Otub2^{-/-}$ mice was analyzed by Western blot (left panel). The right panel shows the relative quantification normalized to GAPDH. Unpaired Student's t test, $n = 6$/group, biological replicates. (C, D) Brains of 8-week-old male $Otub2^{+/+}$ and $Otub2^{-/-}$ mice were analyzed by H&E (C) and Nissl (D) staining. (E) After MCAO surgery, cerebral infarct size was analyzed by TTC staining. (F) Cerebral infarct volume was calculated based on TTC staining. Two-way ANOVA, $n = 5$/group, biological replicates. (G–K) Neurological function was evaluated by mNSS test (G), rotarod test (H), corner turning test (I), and adhesive removal test (J, K). Two-way ANOVA, $n = 5$–8/group, biological replicates. Data in (B, F, G–K) show the mean ± SEM. Source data are available online for this figure.

infarct volume was significantly reduced in $Otub2^{-/-}$ mice on day 1 after cerebral ischemia (Fig. 3M,N).

To directly confirm that OTUB2 enhances neuronal necroptosis after the ischemic insult, $Otub2^{+/+}$ and $Otub2^{-/-}$ HT22 cells were subjected to oxygen-glucose deprivation (OGD). OTUB2 deficiency

reduced OGD-induced cell death and the inhibition of necroptosis, rather than apoptosis, blunted the difference in cell viability between the two genotypes (Fig. 3O; Appendix Fig. S4F), demonstrating that OTUB2 potentiates OGD-induced neuronal necroptosis. Moreover, OGD-induced phosphorylation of RIPK3 and MLKL was strongly

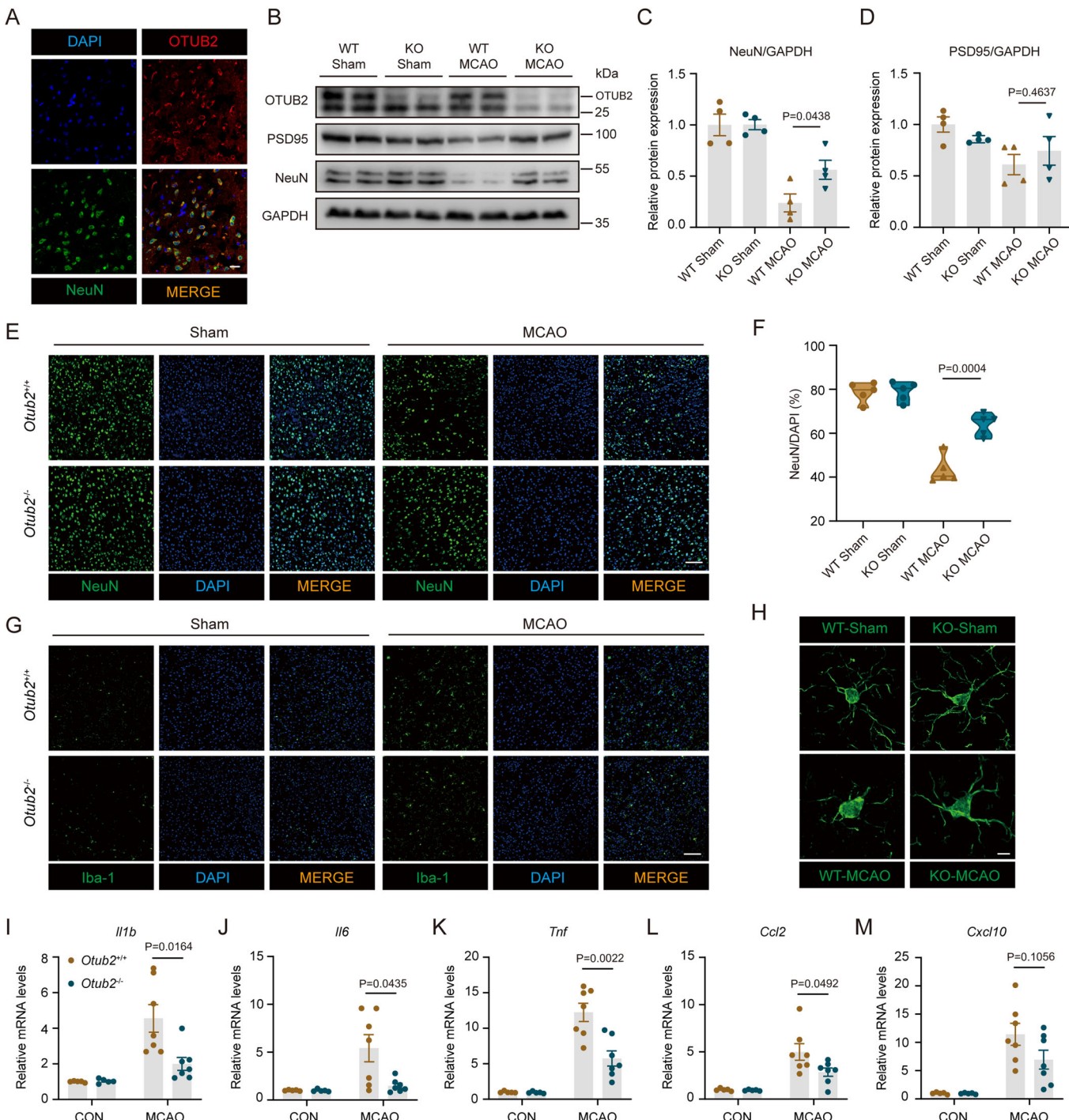

**Figure 2. OTUB2 deficiency diminishes MCAO-induced neuronal loss and neuroinflammation.**

(A) Representative immunofluorescence staining of OTUB2 (red) and NeuN (green) in the brain of C57BL/6 mice. Scale bar, 20 μm. (B) On day 2 after MCAO, the ischemic cerebral hemisphere of $Otub2^{+/+}$ and $Otub2^{-/-}$ mice was analyzed by Western blot with indicated antibodies. (C, D) The relative expression of NeuN (C) and PSD95 (D) was quantified after normalization to GAPDH. Unpaired Student's t test, $n = 4$/group, biological replicates. (E, F) Representative immunofluorescence staining (E) and quantification (F) of NeuN$^+$ cells in the ischemic penumbra on day 2 after MCAO. Scale bar, 100 μm. Unpaired Student's t test, $n = 5$/group, biological replicates. (G) Representative immunofluorescence staining of Iba1$^+$ cells in the ischemic penumbra on day 2 after MCAO. Scale bar, 100 μm. (H) Representative z-stack images of Iba1$^+$ cells in the ischemic penumbra on day 2 after MCAO. Scale bar, 5 μm. (I–M) Forty-eight hours after MCAO, the relative transcription of $Il1b$ (I), $Il6$ (J), $Tnf$ (K), $Ccl2$ (L), and $Cxcl10$ (M) in the ischemic cerebral hemisphere was determined by qRT-PCR. Mann–Whitney U test (I) and Unpaired Student's t test (J–M), $n = 5–7$/group, biological replicates. Data in (C, D, I–M) show the mean ± SEM. Source data are available online for this figure.

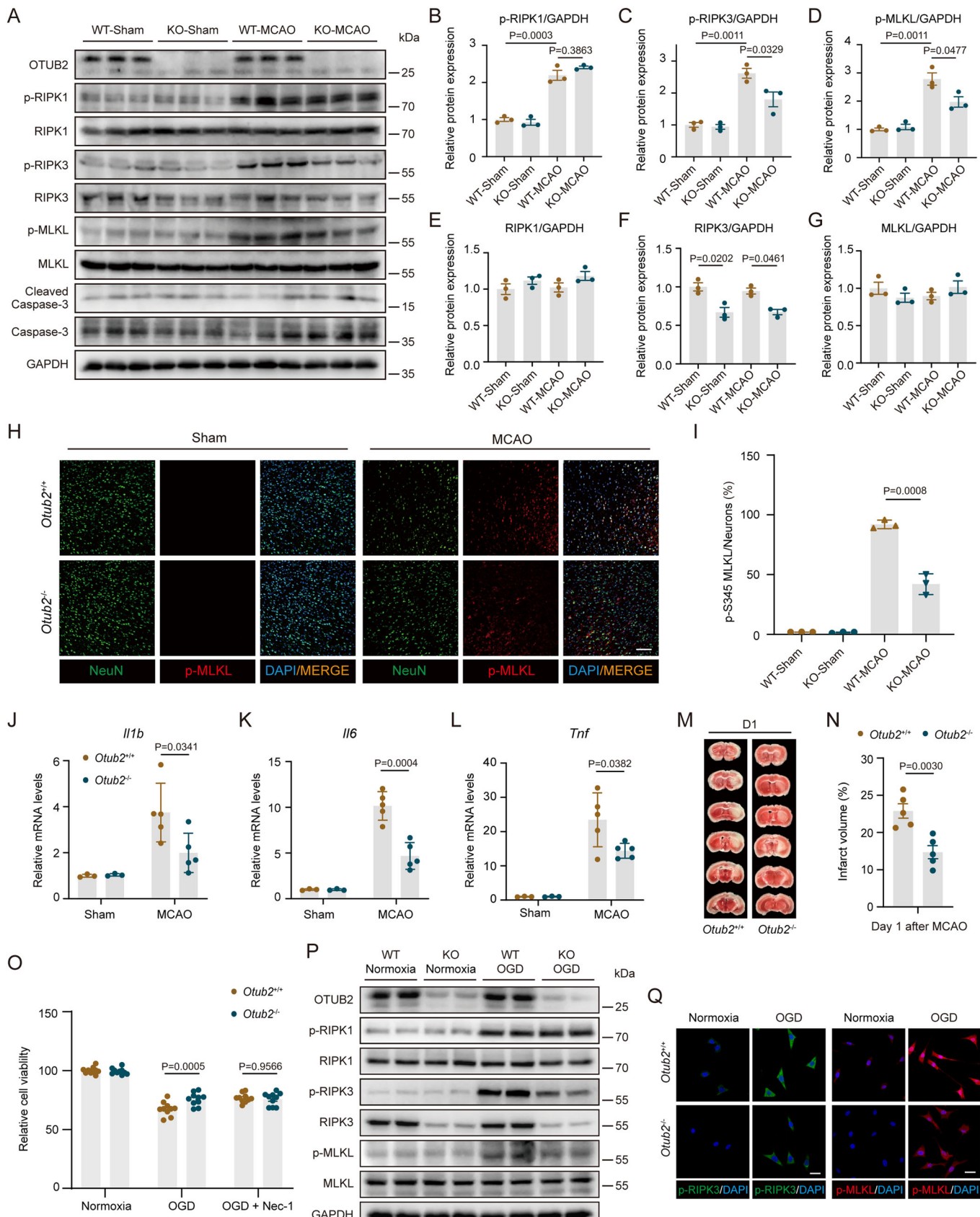

**Figure 3. OTUB2 deficiency ameliorates RIPK3-mediated neuronal necroptosis.**

(A–G) Four hours after MCAO, the ischemic cerebral hemisphere was analyzed by Western blot with indicated antibodies. Representative immunoblots (A) as well as relative quantification of p-RIPK1 (B), p-RIPK3 (C), p-MLKL (D), RIPK1 (E), RIPK3 (F), and MLKL (G) are shown. Two-way ANOVA, $n = 3$/group, biological replicates. (H, I) Four hours after MCAO, the ischemic penumbra was analyzed by immunofluorescence with indicated antibodies. Representative images (H) and percentages of p-MLKL$^+$ neurons (I) are shown. Scale bar, 100 μm. Unpaired Student's t test, $n = 3$/group, biological replicates. (J–L) Four hours after MCAO, the relative transcription of $Il1b$ (J), $Il6$ (K), and $Tnf$ (L) in the ischemic cerebral hemisphere was determined by qRT-PCR. Unpaired Student's t test, $n = 3$–5/group, biological replicates. (M, N) Representative TTC staining (M) and percentages of cerebral infarct volume (N) on day 1 after MCAO. Unpaired Student's t test, $n = 5$/group, biological replicates. (O) $Otub2^{+/+}$ and $Otub2^{-/-}$ HT22 cells were subjected to OGD for 6 h followed by reoxygenation for 12 h in the presence or absence of Nec-1 (50 μM). Cell viability was measured by CCK-8 test. Two-way ANOVA, $n = 10$/group, biological replicates. (P) $Otub2^{+/+}$ and $Otub2^{-/-}$ HT22 cells were subjected to OGD for 6 h followed by reoxygenation for 12 h. Whole-cell lysates were analyzed by Western blot with indicated antibodies. (Q) After OGD treatment for 6 h followed by reoxygenation for 12 h, $Otub2^{+/+}$ and $Otub2^{-/-}$ HT22 cells were analyzed by immunofluorescence with indicated antibodies. Scale bar, 20 μm. Data in (B–G, I, J–L, N, O) show the mean ± SEM. Source data are available online for this figure.

reduced in $Otub2^{-/-}$ HT22 cells while RIPK1 phosphorylation was not affected (Fig. 3P,Q; Appendix Fig. S5A,B). Importantly, OTUB2 deletion reduced RIPK3 protein levels in HT22 cells (Fig. 3P), consolidating the in vivo findings (Fig. 3A). Collectively, these results show that OTUB2 participates in ischemic stroke injury by enhancing RIPK3-mediated neuronal necroptosis.

## OTUB2 physically interacts with RIPK3 to inhibit its degradation

Although OTUB2 deletion significantly reduced RIPK3 protein levels in HT22 cells (Fig. 4A,B), OTUB2 deficiency did not change the transcription of $Ripk3$ mRNA (Fig. 4C), indicating that OTUB2 regulates RIPK3 at post-translational levels. In contrast, OTUB2 deficiency did not alter RIPK1 expression at both protein and mRNA levels in HT22 cells (Appendix Fig. S6A–C). Similar to HT22 cells, primary neurons also had reduced RIPK3 protein levels in the absence of OTUB2 (Fig. 4D). Consistently, overexpression of OTUB2 increased RIPK3 protein levels but did not change RIPK1 expression (Fig. 4E; Appendix Fig. S6D). In line with these in vitro results, in the mouse brain, OTUB2 deficiency reduced RIPK3 only at protein levels but not mRNA levels, whereas both protein and mRNA levels of RIPK1 were not altered by OTUB2 deletion (Appendix Fig. S6E–I).

We then investigated whether OTUB2 regulated RIPK3 by direct interaction. Co-immunoprecipitation (Co-IP) verified the interaction between endogenous RIPK3 and OTUB2 in HT22 cells (Fig. 4F). Besides, we also confirmed the direct interaction between exogenously expressed FLAG-OTUB2 and GFP-RIPK3 by Co-IP (Fig. 4G,H). Moreover, OTUB2 and RIPK3 were found to co-localize in the cytoplasm of HT22 cells and primary neurons, as revealed by immunofluorescence staining (Fig. 4I; Appendix Fig. S7). Functionally, deletion of OTUB2 accelerated the degradation of RIPK3 while OTUB2 overexpression delayed RIPK3 degradation (Fig. 4J,K; Appendix Fig. S8). Together, these results demonstrate that OTUB2 interacts with RIPK3 and increases the protein abundance of RIPK3 by reducing its degradation.

## OTUB2 inhibits the proteasomal degradation of RIPK3 via K48 deubiquitination

RIPK3 can be degraded by both lysosomes and proteasomes, two proteolytic machineries that degrade ubiquitinated proteins in the cell (Choi et al, 2018; Liu et al, 2022; Mei et al, 2021; Seo et al, 2016). Inhibition of proteasomes rather than lysosomes blunted the

difference in RIPK3 abundance between OTUB2-sufficient and -deficient cells (Fig. 5A,B), suggesting that OTUB2 primarily interferes with proteasome-mediated degradation of RIPK3. Indeed, increased co-localization of RIPK3 with proteasomes was observed in OTUB2-deficient cells compared with OTUB2-sufficient cells (Fig. 5C). In comparison, OTUB2 deficiency had no impact on the co-localization of RIPK3 with lysosomes (Appendix Fig. S9). Given that the proteasomal degradation of RIPK3 requires K48 ubiquitination (Choi et al, 2018; Mei et al, 2021), we analyzed the effect of OTUB2 on RIPK3 ubiquitination. OTUB2 overexpression reduced the total and K48-specific ubiquitination of RIPK3 (Fig. 5D,E). However, OTUB2 had no impact on the K11, K33, and K63-specific ubiquitination of RIPK3 (Appendix Fig. S10A–C), indicating that OTUB2 specifically reduces the K48 ubiquitination of RIPK3. Consistently, ablation of OTUB2 increased the K48 ubiquitination of RIPK3 in the brain (Fig. 5F,G). Moreover, the in vitro deubiquitination assay validated that OTUB2 could directly remove K48-linked polyubiquitin chains that were conjugated on RIPK3 (Fig. 5H). Notably, the DUB activity of OTUB2 is mediated by the C51 residue (Fig. 5I). The OTUB2 C51S mutant, which was deprived of the enzymatic activity, failed to K48 deubiquitinate RIPK3 (Fig. 5J). Concordantly, the C51S mutant also lost the ability to upregulate RIPK3 abundance (Fig. 5K). In aggregate, these data show that OTUB2 reduces K48-specific ubiquitination on RIPK3 via the deubiquitinating activity, thereby inhibiting the proteasomal degradation of RIPK3.

## OTUB2 enhances neuronal necroptosis through RIPK3

Given that RIPK3 critically regulates necroptosis and that OTUB2 increases RIPK3 abundance and activity, we then assessed whether OTUB2 enhanced neuronal necroptosis via regulating RIPK3. Compared with $Otub2^{+/+}$ primary neurons, $Otub2^{-/-}$ primary neurons died significantly less upon OGD treatment (Fig. 6A,B; Appendix Fig. S11A–C). Besides, OTUB2 deficiency significantly reduced the levels of phosphorylated RIPK3 and MLKL in primary neurons after OGD (Fig. 6C–F), showing that OTUB2 increases neuronal necroptosis after ischemia. Next, we specifically induced necroptosis with a cocktail of TNF-α, CHX, and Z-VAD-FMK (TCZ) and found that $Otub2^{-/-}$ cells died significantly less than $Otub2^{+/+}$ cells (Fig. 6G; Appendix Fig. S11D), unambiguously demonstrating that OTUB2 deletion diminishes necroptosis. Treatment with the RIPK3 inhibitor GSK-872 or the MLKL inhibitor GW806742X restored the viability of $Otub2^{+/+}$ cells and erased the difference between the two genotypes after TCZ stimulation (Fig. 6G; Appendix Fig. S11E), indicating that OTUB2

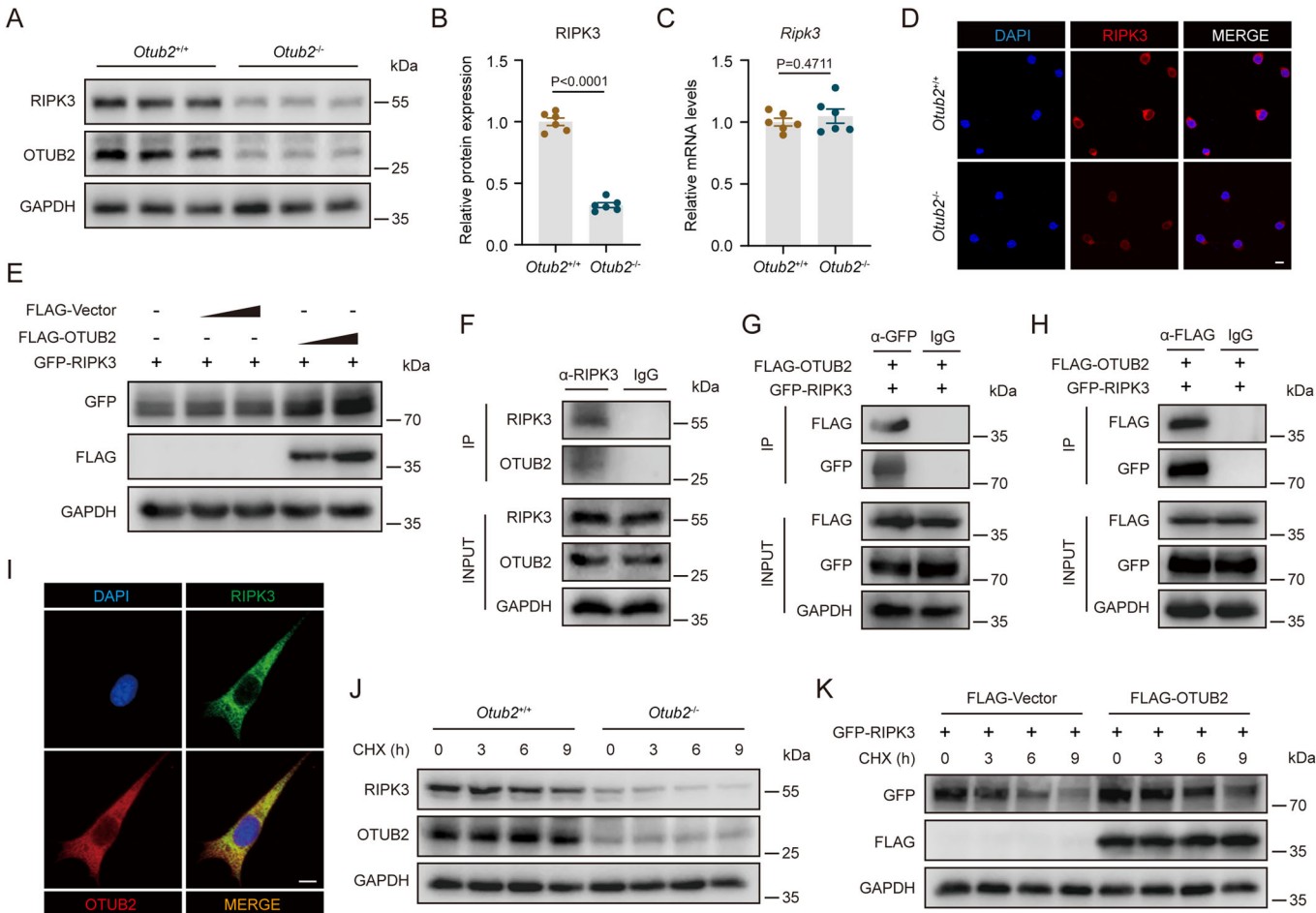

**Figure 4. OTUB2 interacts with RIPK3 and reduces its degradation.**

(A, B) Whole-cell lysates of *Otub2*[+/+] and *Otub2*[−/−] HT22 cells were analyzed by Western blot with indicated antibodies. Representative immunoblots (A) and relative RIPK3 quantification (B) are shown. Unpaired Student's t test, n = 6/group, biological replicates. (C) Relative *Ripk3* mRNA levels in *Otub2*[+/+] and *Otub2*[−/−] HT22 cells were determined by qRT-PCR. Unpaired Student's t test, n = 6/group, biological replicates. (D) RIPK3 abundance in primary neurons from *Otub2*[+/+] and *Otub2*[−/−] mice were analyzed by immunofluorescence. Scale bar, 10 μm. (E) GFP-RIPK3 plasmids were co-transfected into NIH/3T3 cells with increasing amount of FLAG-Vector or FLAG-OTUB2 plasmids for 24 h. Whole-cell lysates were analyzed by Western blot. (F) Immunocomplexes were harvested from HT22 whole-cell lysates by immunoprecipitation and then analyzed by Western blot with indicated antibodies. (G, H) FLAG-OTUB2 and GFP-RIPK3 plasmids were co-transfected into NIH/3T3 cells for 24 h. Immunocomplexes were harvested from whole-cell lysates by immunoprecipitation with anti-GFP (G) or anti-FLAG (H) antibodies, followed by Western blot analysis. (I) Subcellular distribution of RIPK3 (green) and OTUB2 (red) in HT22 cells was determined by immunofluorescence. Scale bar, 10 μm. (J) After treatment with CHX (50 μM) for indicated time, *Otub2*[+/+] and *Otub2*[−/−] HT22 cells were lysed and analyzed by Western blot with indicated antibodies. (K) NIH/3T3 cells were transfected with indicated plasmids for 24 h and then treated with CHX (50 μM) for indicated time before lysis. Whole-cell lysates were analyzed by Western blot. Data in (B, C) show the mean ± SEM. Source data are available online for this figure.

promotes necrosis by regulating the RIPK3-MLKL signaling. Moreover, overexpression of OTUB2 increased necroptosis whereas overexpression of the OTUB2 C51S mutant exerted no effect (Fig. 6H). Therefore, the necroptosis-enhancing function of OTUB2 is critically dependent on its catalytic activity, which is indispensable for deubiquitinating and stabilizing RIPK3. Collectively, these findings show that OTUB2 enhances ischemia-induced neuronal necroptosis by regulating RIPK3 through its DUB activity.

## Pharmacological inhibition of OTUB2 ameliorates ischemic stroke injury

In line with our in vitro findings (Fig. 6), pharmacological inhibition of RIPK3 with GSK-872 significantly reduced cerebral

infarct volume and neurological deficits in *Otub2*[+/+] mice after MCAO and abrogated the difference between *Otub2*[+/+] and *Otub2*[−/−] strains (Fig. 7A–D; Appendix Fig. S12A–D). Besides, *Otub2*[+/+] and *Otub2*[−/−] mice had comparable amounts of phosphorylated RIPK3 and MLKL in the ischemic brain after GSK-872 treatment (Fig. 7E; Appendix Fig. S13A). These in vitro and in vivo data synergistically demonstrate that OTUB2 enhances ischemic stroke injury by potentiating neuronal necroptosis through RIPK3.

Since OTUB2 promoted neuronal necroptosis and genetic deletion of OTUB2 ameliorated ischemic stroke injury, we further investigated whether inhibition of OTUB2 with inhibitors could protect mice from ischemic stroke. LN5P45 is a small-molecule inhibitor of OTUB2 with high specificity, and it inhibits the DUB

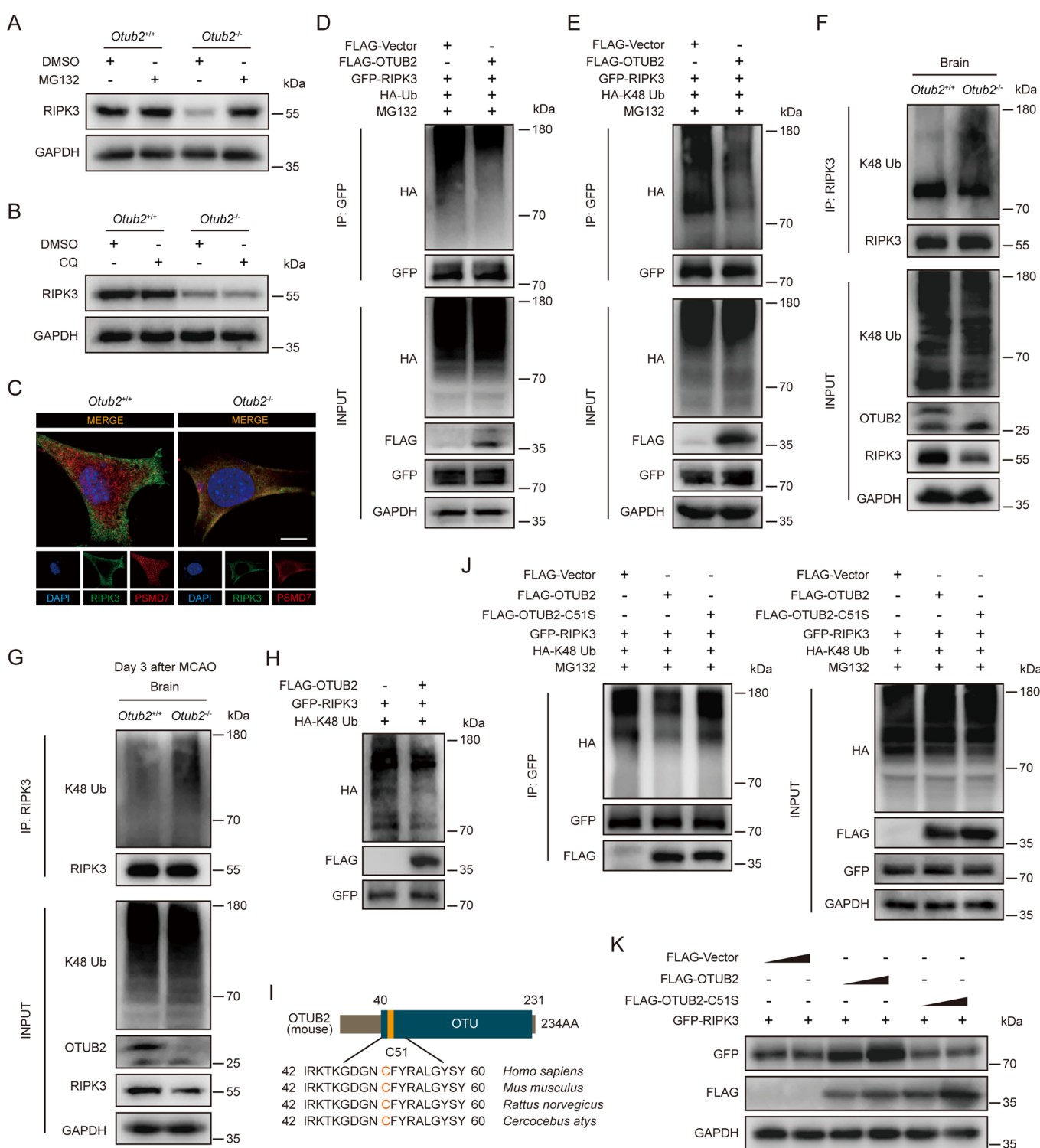

activity of OTUB2 by engaging the C51 active site (Appendix Fig. S13B) (Gan et al, 2023). We found that LN5P45 treatment decreased RIPK3 protein levels in HT22 cells in a dose-dependent manner (Appendix Fig. S13C). Mechanistically, LN5P45 induced the degradation of RIPK3 by increasing the K48 ubiquitination of RIPK3 (Appendix Fig. S13D). LN5P45 treatment restored the viability of $Otub2^{+/+}$ cells to that of $Otub2^{-/-}$ cells in response to

TCZ stimulation (Appendix Fig. S13E). Similar to GSK-872, LN5P45 could also cross the blood-brain barrier (BBB) (Appendix Figs. S12A–D and S14A–C). Intriguingly, administration of LN5P45 significantly reduced cerebral infarction and neurological deficits in $Otub2^{+/+}$ mice accompanied with decreased activation of RIPK3 and MLKL, yielding similar therapeutic effects as GSK-872 (Fig. 7F–J; Appendix Fig. S13A).

**Figure 5. OTUB2 removes K48 ubiquitination from RIPK3 through the C51 active site.**

(A, B) After treatment with 5 µM of MG132 (A) or 10 µM of CQ (B) for 6 h, *Otub2*<sup>+/+</sup> and *Otub2*<sup>−/−</sup> HT22 cells were lysed and analyzed by Western blot with indicated antibodies. (C) Subcellular distribution of PSMD7 (red) and RIPK3 (green) in HT22 cells was determined by immunofluorescence. Scale bar, 10 µm. (D, E) NIH/3T3 cells were transfected with indicated plasmids for 24 h, followed by treatment with MG132 (5 µM) for 6 h. Immunocomplexes were harvested from whole-cell lysates by immunoprecipitation and analyzed by Western blot. (F) Brains of untreated *Otub2*<sup>+/+</sup> and *Otub2*<sup>−/−</sup> mice were lysed for protein extraction. Immunocomplexes were harvested from brain lysates and analyzed by Western blot. (G) On day 3 after MCAO, brains of *Otub2*<sup>+/+</sup> and *Otub2*<sup>−/−</sup> mice were lysed for protein extraction. Immunocomplexes were harvested from brain lysates and analyzed by Western blot. (H) Representative immunoblots of the in vitro deubiquitination assay. (I) Schematic diagram of the active residue (C51) in OTUB2. (J) NIH/3T3 cells were transfected with indicated plasmids for 24 h, followed by treatment with MG132 (5 µM) for 6 h. Cell lysates were subjected to immunoprecipitation and analyzed by Western blot. (K) NIH/3T3 cells were transfected with indicated plasmids for 24 h before lysis. Whole-cell lysates were analyzed by Western blot with indicated antibodies. Source data are available online for this figure.

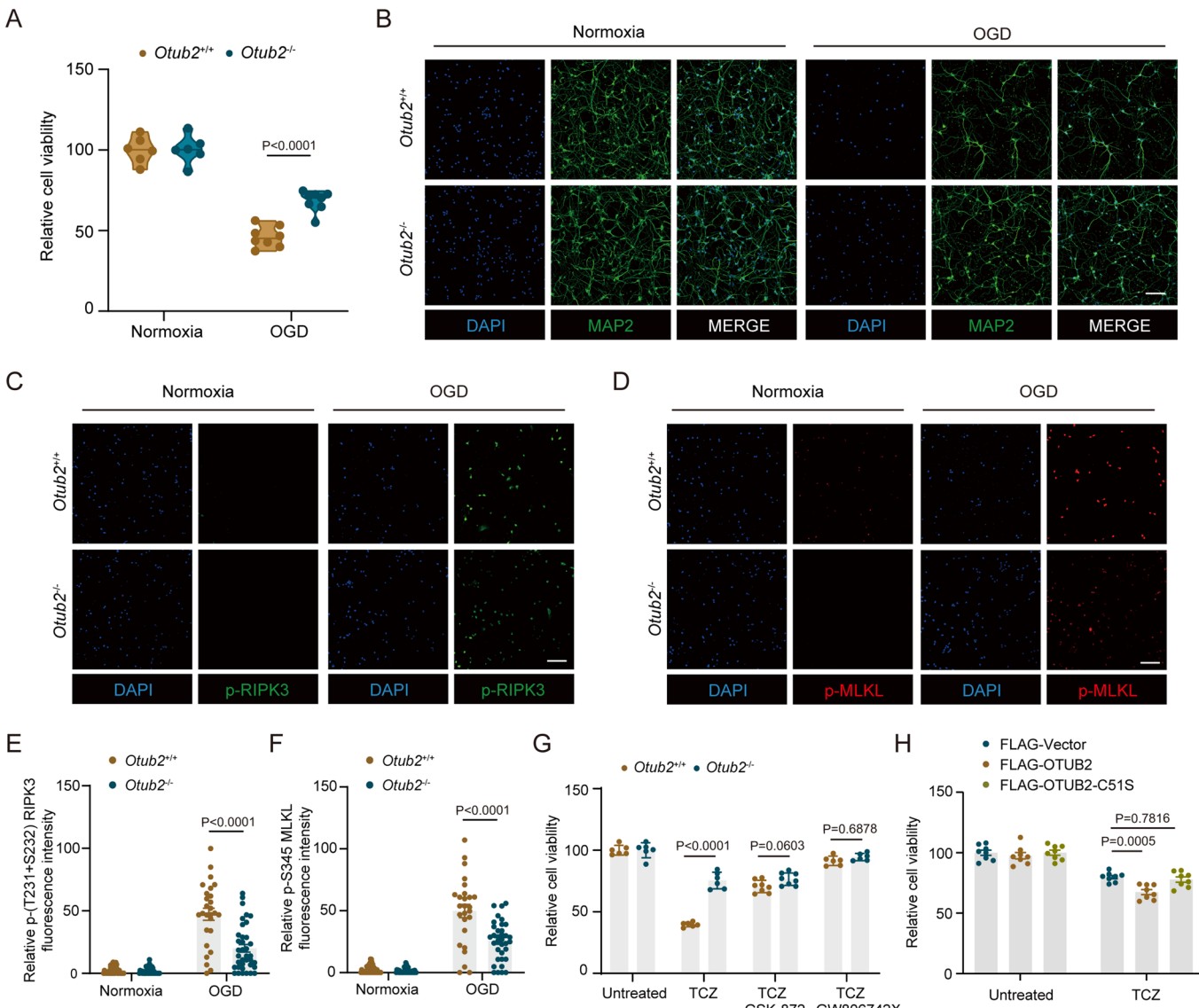

**Figure 6. OTUB2 potentiates neuronal necroptosis by regulating RIPK3.**

(A) Primary neurons from *Otub2*<sup>+/+</sup> and *Otub2*<sup>−/−</sup> mice were subjected to OGD treatment for 6 h followed by reoxygenation for 12 h. Cell viability was determined by CCK-8 test. Unpaired Student's t test, $n = 6$–8/group, biological replicates. (B) After OGD treatment for 6 h followed by reoxygenation for 12 h, primary neurons from *Otub2*<sup>+/+</sup> and *Otub2*<sup>−/−</sup> mice were analyzed by immunofluorescence with anti-MAP2 antibody. Scale bar, 100 µm. (C–F) After OGD treatment, primary neurons were analyzed by immunofluorescence with indicated antibodies. Representative p-RIPK3 (C) and p-MLKL (D) staining as well as relative p-RIPK3 (E) and p-MLKL (F) fluorescence intensity are shown. Scale bar, 100 µm. Mann–Whitney U test (E) and unpaired Student's t test (F), $n = 25$–50/group, biological replicates. (G) *Otub2*<sup>+/+</sup> and *Otub2*<sup>−/−</sup> HT22 cells were left untreated or treated with TNF-α (20 ng/ml) + CHX (50 µM) + Z-VAD-FMK (20 µM) (TCZ) in the presence or absence of GSK-872 (25 µM) or GW806742X (1 µM) for 6 h. Cell viability was determined by CCK-8 test. Two-way ANOVA, $n = 6$–8/group, biological replicates. (H) HT22 cells were transfected with indicated plasmids for 24 h, followed by TCZ treatment. Cell viability was determined by CCK-8 test. Two-way ANOVA, $n = 8$/group, biological replicates. Data in (E–H) show the mean ± SEM. Source data are available online for this figure.

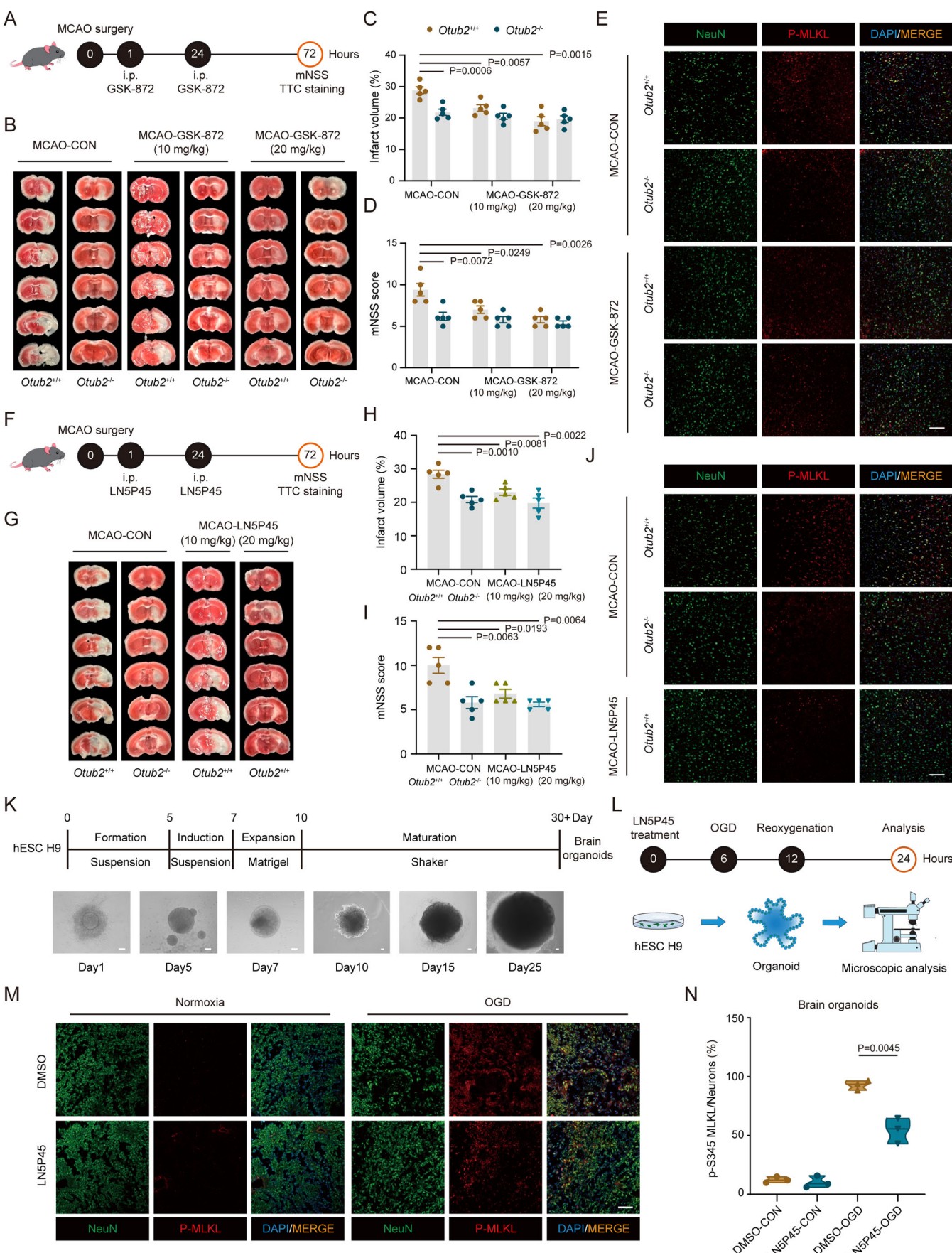

**Figure 7. OTUB2 exacerbates ischemic stroke injury by regulating RIPK3.**

(A) Experimental flowchart for GSK-872 administration. (B, C) Representative TTC staining (B) and percentages of cerebral infarct volume (C) on day 3 after MCAO. Multiple Student's t tests, $n = 5$/group, biological replicates. (D) On day 3 after MCAO, neurological functions were assessed by mNSS scores. Multiple Student's t tests, $n = 5$/group, biological replicates. (E) Mice were intraperitoneally injected with GSK-872 (10 mg/kg) prior to MCAO surgery. The ischemic penumbra was analyzed by immunofluorescence at 6 h after MCAO. Scale bar, 100 μm. (F) Experimental flowchart for LN5P45 administration. (G, H) Representative TTC staining (G) and percentages of cerebral infarct volume (H) on day 3 after MCAO. Multiple Student's t tests, $n = 5$/group, biological replicates. (I) On day 3 after MCAO, neurological functions were assessed by mNSS scores. Mann–Whitney U test, $n = 5$/group, biological replicates. (J) Mice were intraperitoneally injected with LN5P45 (10 mg/kg) prior to MCAO surgery. The ischemic penumbra was analyzed by immunofluorescence at 6 h after MCAO. Scale bar, 100 μm. (K) Schematic diagram describing the generation of human brain organoids. The lower panel shows different stages of brain organoids. Scale bar, 100 μm. (L) Experimental flowchart for the analysis of brain organoids. After 30 days of culture, brain organoids were treated as described in the flowchart before analysis. (M, N) After OGD, the brain organoids were analyzed by immunofluorescence with indicated antibodies. Representative images (M) and percentages of p-MLKL$^+$ neurons (N) are shown. Scale bar, 50 μm. Unpaired Student's t test, $n = 3$/group, biological replicates. Data in (C, D, H, I) show the mean ± SEM. Source data are available online for this figure.

To test the potential of OTUB2 inhibition in the clinical treatment for ischemic stroke, we examined the effect of LN5P45 on OGD-induced neuronal death in human brain organoids (Fig. 7K,L). LN5P45 treatment significantly reduced OGD-induced MLKL phosphorylation and neuronal loss in brain organoids (Fig. 7M,N; Appendix Fig. S15), indicating that OTUB2 may become a druggable target for the treatment of ischemic stroke. Taken together, these results demonstrate that OTUB2 aggravates ischemic brain injury through RIPK3, and pharmacological inhibition of OTUB2 ameliorates cerebral injury after ischemic stroke.

## Discussion

After ischemic stroke, key pathological processes, such as neuronal death and neuroinflammation, are tightly regulated by ubiquitin-modifying enzymes (Huang et al, 2024). In the present study, we found that the DUB OTUB2 enhanced cerebral ischemic injury by increasing RIPK3-mediated neuronal necroptosis. Moreover, genetic deletion or pharmacological inhibition of OTUB2 significantly ameliorated ischemic stroke injury, suggesting that OTUB2 inhibition may serve as a potential therapeutic approach for ischemic stroke.

Neuronal necroptosis is a core event after cerebral ischemia (Degterev et al, 2005; Naito et al, 2020). Necroptosis is critically regulated by RIPK1, RIPK3, TIR domain-containing adaptor-inducing IFNβ (TRIF), and Z-DNA-binding protein 1 (ZBP1), all of which contain the RIP homotypic interaction motif (RHIM) domain (Newton et al, 2024; Yuan and Ofengeim, 2024). TNF receptor (TNFR) family receptors, Toll-like receptors (TLRs), and Z-nucleic acids induce the binding and activation of RIPK1, TRIF, and ZBP1, respectively (Ai et al, 2024). Subsequently, activated RIPK1, TRIF, and ZBP1 interact with RIPK3 through RHIM engagement, leading to RIPK3 activation and necroptosis (Jiao et al, 2020; Kaiser et al, 2013; Li et al, 2012). Thus, RIPK3 serves as a signal transduction center for various necroptotic signaling pathways. In the context of cerebral ischemia, which induces the RIPK1-RIPK3-MLKL necroptotic pathway, we found that OTUB2 specifically regulated RIPK3, but not RIPK1, to impinge on neuronal necroptosis. Of note, ubiquitination is a key mechanism orchestrating the optimized expression of RIPK3. Multiple E3 ligases, including PELI1, TRIM11, TRIM25, and CHIP (Choi et al, 2018; Mei et al, 2021; Seo et al, 2016; Xie et al, 2020), can ubiquitinate RIPK3 for degradation in the proteasome or

lysosome. In the current study, we found that OTUB2 removed K48-specific polyubiquitination from RIPK3 and thereby inhibited its proteasomal degradation, which is consistent with previous reports that RIPK3 is degraded by the proteasome after K48 ubiquitination (Choi et al, 2018; Mei et al, 2021). Importantly, to the best of our knowledge, OTUB2 is the first DUB that has been shown to stabilize RIPK3, enriching the regulatory mechanism for necroptosis.

After cerebral ischemia, intracellular contents are released from necroptotic neurons through the ruptured membrane (Huang et al, 2024). Leaked intracellular contents, such as high-mobility group box 1 protein (HMGB1) and S100 proteins (S100s), subsequently induce inflammatory responses and they are classified as damage-associated molecular patterns (DAMPs) (Gong et al, 2020). In agreement with the well-established concept that necroptosis contributes to inflammation (Pasparakis and Vandenabeele, 2015), we observed that the impaired necroptosis in $Otub2^{-/-}$ mice was associated with decreased inflammation. However, given that inflammatory signal transduction is tightly controlled by ubiquitination (Huang et al, 2024; Ruan et al, 2022), it is also possible that OTUB2 directly interferes with inflammatory responses. For example, the E3 ligase TRIM45 and the DUB USP25 regulate ischemic stroke injury in mice by increasing and decreasing microglia-mediated neuroinflammation, respectively (Li et al, 2023; Xia et al, 2022). However, in response to LPS, which stimulates the same pattern recognition receptors as typical DAMPs released from necroptotic neurons, cytokine production in microglia was not affected by OTUB2 deletion. Therefore, the diminished neuroinflammation in $Otub2^{-/-}$ mice should be attributed to reduced DAMP leakage from necroptotic neurons rather than direct OTUB2 regulation, further corroborating the necroptosis-promoting role of OTUB2.

The DUB activity of OTUB2 is exerted by its C51 catalytic residue (Ren et al, 2024; Zhang et al, 2019). However, OTUB1, a DUB that shares structural domains and sequences with OTUB2, regulates many protein substrates independent of its enzymatic activity (Wang et al, 2019; Zhao et al, 2023). We found that the enzymatically inactive OTUB2 C51S mutant failed to deubiquitinate RIPK3, demonstrating that OTUB2 stabilizes RIPK3 through its enzymatic activity. A recently discovered OTUB2 inhibitor LN5P45 forms a covalent bond with the C51 site of OTUB2, thereby sabotaging OTUB2 catalytic activity (Gan et al, 2023). LN5P45 treatment reduced RIPK3 protein levels by increasing the K48 ubiquitination of RIPK3, further confirming that OTUB2 deubiquitinates RIPK3 through the enzymatic activity.

Pharmacological inhibition of OTUB2 by LN5P45 significantly reduced ischemic stroke injury in mice, indicating that specific and potent OTUB2 inhibitors are competitive candidates for drugs treating ischemic stroke. Besides, the global deletion of OTUB2 had no obvious impact on mice under physiological conditions, implying that OTUB2 inhibition may not produce deleterious side effects. In summary, our findings identified OTUB2 as a key regulator of cerebral ischemic injury and elucidated the mechanism of OTUB2 in neuronal necroptosis, providing a potential therapeutic target for ischemic stroke.

# Methods

### Reagents and tools table

| Reagent/Resource | Reference or Source | Identifier or Catalog Number |
|---|---|---|
| **Experimental models** | | |
| C57BL/6 (*M. musculus*) | Cyagen Biosciences | |
| *Otub2*$^{+/-}$ (*M. musculus*) | Cyagen Biosciences | |
| *Otub2*$^{-/-}$ (*M. musculus*) | Cyagen Biosciences | |
| Human embryonic stem cells (hESCs; H9 cell line) | National Collection of Authenticated Cell Cultures | |
| HT22 cells | National Collection of Authenticated Cell Cultures | |
| NIH/3T3 cells | National Collection of Authenticated Cell Cultures | |
| *Otub2*$^{-/-}$ HT22 cells | This study | |
| **Plasmids/Vectors** | | |
| FLAG-Vector | GeneChem | |
| FLAG-OTUB2 | GeneChem | |
| FLAG-OTUB2-C51S | GeneChem | |
| GFP-RIPK3 | GeneChem | |
| HA-Ub | GeneChem | |
| HA-K11 Ub | GeneChem | |
| HA-K33 Ub | GeneChem | |
| HA-K48 Ub | GeneChem | |
| HA-K63 Ub | GeneChem | |
| MYC-HIS-RIPK1 | GeneChem | |
| **Antibodies (dilution)** | | |
| NeuN (WB1:1000, IF1:200) | Abcam/Proteintech | ab177487/66836-1-Ig |
| p-MLKL (WB1:1000, IF1:200) | Abcam | ab196436 |
| Caspase-3 (WB1:1000) | Cell Signaling Technology | 14220S |
| Iba-1 (IF1:200) | Cell Signaling Technology | 17198S |
| K48-linkage Specific Polyubiquitin (WB1:1000) | Cell Signaling Technology | 8081S |

| Reagent/Resource | Reference or Source | Identifier or Catalog Number |
|---|---|---|
| p-RIPK1 (WB1:1000) | Cell Signaling Technology | 31122S |
| p-RIPK3 (WB1:1000, IF1:200) | Cell Signaling Technology | 91702S |
| RIPK1 (WB1:1000) | Cell Signaling Technology | 3493T |
| RIPK3 (WB1:1000, IF1:200) | Cell Signaling Technology | 95702S |
| OTUB2 (WB1:1000, IF1:200) | Novus Biologicals | NBP2-03223 |
| FLAG tag (WB1:1000) | Proteintech | 20543-1-AP |
| GAPDH (WB1:10000) | Proteintech | 60004-1-Ig |
| GFP tag (WB1:1000) | Proteintech | 50430-2AP |
| HA tag (WB1:1000) | Proteintech | 51064-2-AP |
| LAMP1 (IF1:200) | Proteintech | 21997-1-AP |
| MAP2 (IF1:200) | Proteintech | 67015-1-Ig |
| MLKL (WB1:1000) | Proteintech | 66675-1-Ig |
| MYC tag (WB1:1000) | Proteintech | 60003-2-Ig |
| PSD95 (WB1:1000) | Proteintech | 20665-1-AP |
| PSMD7 (IF1:200) | Proteintech | 16034-1-AP |
| **Oligonucleotides and other sequence-based reagents** | | |
| Primers for quantitative Real-Time PCR (qRT-PCR) | This study | Appendix Table S1 |
| **Chemicals, Enzymes and other reagents** | | |
| Phosphatase Inhibitor Cocktail | APE×BIO | K1015 |
| Protease Inhibitor Cocktail | APE×BIO | K4002 |
| Nissl Staining | Beyotime | C0117 |
| BeyoMag™ Protein A + G Magnetic Beads | Beyotime | P2108 |
| L-Glutamine | Biosharp | BS179 |
| Poly-L-lysine | Biosharp | BL556A |
| Quick Start™ Bradford Protein Assay Kit | BIO-RAD | 5000201 |
| Protein Extraction Reagents | Boster Bio | AR0101/AR0103 |
| Corning® Matrigel® | Corning | 354277 |
| 5× DualColor Protein Loading Buffer | FUDE Biological Technology | FD006 |
| B27 Minus Insulin | Gibco | A1895601 |
| DMEM medium | Gibco | 11965118 |
| Glucose-Free DMEM | Gibco | 11966025 |
| Neurobasal™ Medium | Gibco | 21103049 |
| Opti-MEM™ | Gibco | 31985070 |
| 0.25% Trypsin | Gibco | 25200072 |
| Chloroquine | MedChemExpress | HY-17589A |
| Corn oil | MedChemExpress | HY-Y1888 |
| GSK-872 | MedChemExpress | HY-101872 |
| GW806742X | MedChemExpress | HY-112292 |

| Reagent/Resource | Reference or Source | Identifier or Catalog Number |
|---|---|---|
| LN5P45 | MedChemExpress | HY-149482 |
| MG132 | MedChemExpress | HY-13259 |
| Nec-1 | MedChemExpress | HY-15760 |
| Z-VAD-FMK | MedChemExpress | HY-16658B |
| Cycloheximide | Merck | 239763-M |
| Ncm ECL Ultra | New Cell & Molecular Biotech | P10300 |
| TNF-α | PeproTech | 315-01A |
| TUNEL staining kit | Proteintech | PF00006 |
| Cytosine arabinoside | Sigma-Aldrich | C1768 |
| 2, 3, 5-triphenyltetrazolium chloride (TTC) | Sigma-Aldrich | T8877 |
| 5% BSA Blocking Buffer | Solarbio | SW3015 |
| DMSO | Solarbio | D8371 |
| Hematoxylin-Eosin (H&E) | Solarbio | G1120 |
| Paraformaldehyde | Solarbio | P1110 |
| Penicillin-Streptomycin Liquid | Solarbio | P1400 |
| 0.5% Triton X-100 | Solarbio | T8200 |
| Gentle Cell Dissociation Reagent | STEMCELL Technologies | 100-0485 |
| mTeSR™ medium | STEMCELL Technologies | 85850 |
| STEMdiff™ Cerebral Organoid Kit | STEMCELL Technologies | 08570 |
| Y-27632 | STEMCELL Technologies | 72302 |
| Brain Slicing Mold | RWD life science | 68707 |
| Isoflurane | RWD life science | R510-22-10 |
| MCAO sutures | RWD life science | MSMC21B120PK50/ MSMC23B120PK50 |
| PrimeScript™ RT reagent Kit | Takara | RR037A |
| TB Green® Premix Ex Taq™ II | Takara | RR820A |
| Lipofectamine 3000 | Thermo Fisher Scientific | L3000015 |
| TRIzol Reagent | Thermo Fisher Scientific | 15596026CN |
| Fetal Bovine Serum | Vazyme | F101-01 |
| **Software** | | |
| MultiQuant™ 3.0.3 software | SCIEX | |
| Evolution Capt | Vilber | |
| GraphPad Prism 8 | GraphPad | |
| ImageJ software | NIH Image | |
| SPSSAU | www.spssau.com | |
| GraphBBB | www.graphbbb.com | |
| **Other** | | |
| Liquid chromatography-triple quadruple mass spectrometer | SCIEX | Qtrap6500 + / Triple Quad™ 4500MD |
| Hypoxic chamber | Billups-Rothenberg | MIC-101 |

| Reagent/Resource | Reference or Source | Identifier or Catalog Number |
|---|---|---|
| Panlab Rotarod | Harvard Apparatus | 76-0770 |
| Forma™ 3 cell incubator | Thermo Fisher Scientific | 4110 |
| Multiskan SkyHigh Microplate Spectrophotometer | Thermo Fisher Scientific | A51119700DPC |
| QuantStudio™ 5 Real-Time PCR System | Thermo Fisher Scientific | Quant Studio™ 5 |
| FV3000 Confocal Laser Scanning Microscope | Olympus | Evident/FV3000 |
| Laser Doppler Perfusion Monitoring | Perimed | PeriFlux 5000/ PeriCam PSI |
| Blood pressure monitor | Softron Biotechnology | BP-2010A |
| Fusion FX.EDGE system | Vilber | |

## Animals

C57BL/6 mice and *Otub2*$^{+/-}$ (C57BL/6 background) mice were provided by Cyagen Biosciences. At the Laboratory Animal Resources Center of Wenzhou Medical University, the mice were housed in a specific-pathogen-free environment with a temperature of 22–25 °C, a humidity of 40–60%, and a 12-h light/12-h dark cycle. Heterozygous *Otub2*$^{+/-}$ mice were bred to produce *Otub2*$^{+/+}$ and *Otub2*$^{-/-}$ mice, and genotyping was performed by PCR. All animal experiments were approved by the Animal Management and Ethics Committee of Wenzhou Medical University (Approval number: wydw2023-0613).

## Cell culture

HT22 cells and NIH/3T3 cells were obtained from the National Collection of Authenticated Cell Cultures (Shanghai, China) and cultured in DMEM medium (Gibco, 11965118) with 10% fetal bovine serum (FBS; Vazyme, F101-01) and 1% Penicillin-Streptomycin Liquid (Solarbio, P1400). The *Otub2*$^{-/-}$ HT22 cell line was generated using the CRISPR/Cas9 gene-editing technology (OTUB2-gRNA: GAACTGCACGATTCGGTCCG) as previously reported (Zhao et al, 2023). Human embryonic stem cells (hESCs; H9 cell line) were obtained from the National Collection of Authenticated Cell Culture and cultured in mTeSR™ medium (STEMCELL Technologies, 85850) in plates coated with the Corning® Matrigel® (Corning, 354277). Cells were detached from plates using the Gentle Cell Dissociation Reagent (STEMCELL Technologies, 100-0485).

Primary glial cells were extracted from neonatal mice and cultured in DMEM containing 10% FBS and 1% Penicillin-Streptomycin until a confluent cell layer was formed. Thereafter, the confluent cell layer of astrocytes was removed by treatment with 0.25% trypsin (Gibco, 25200072). After washing with PBS, the remaining adherent microglia were cultured for further analysis. Primary cortical neurons were isolated from E18.5 embryos. The cortex was digested with 0.25% trypsin, followed by filtration with 70 and 40 μm filters to generate single-cell suspension. Cells were then inoculated in 6-well plates coated with Poly-L-lysine (Biosharp, BL556A) at a density of $7 \times 10^5$ cells/well. Cells were

cultured in Neurobasal Medium (Gibco, 21103049) supplemented with 2% B27 Minus Insulin (Gibco, A1895601), 1% L-Glutamine (Biosharp, BS179), and 1% Penicillin-Streptomycin for 8–9 days, and 5 μM cytosine arabinoside (Sigma-Aldrich, C1768) was added on day 4. All cell cultures were maintained in a Forma™ 3 cell incubator (Thermo Fisher Scientific) with 5% $CO_2$.

## MCAO surgery

Unless otherwise specified, transient MCAO was induced in male mice (8–12 weeks old) as previously reported (Li et al, 2023). In certain experiments, MCAO was performed in 12-month-old male mice to study the impact of OTUB2 on ischemic stroke injury in aged mice. Briefly, a MCAO suture (RWD life science, MSMC21B120PK50/MSMC23B120PK50) was inserted into the left internal carotid artery, and blood flow to the middle cerebral artery was then blocked by the silicone-wrapped tip of the suture. Cerebral blood flow was measured with a laser Doppler monitor (Perimed). MCAO was successfully performed when cerebral blood flow decreased to <25% of baseline. After 70 min of occlusion, the suture was removed to restore cerebral blood flow to >50% of baseline. The sham operation group underwent the same surgery as the MCAO group except for suture insertion.

## Plasmid construction and transfection

The plasmids encoding FLAG-OTUB2, FLAG-OTUB2-C51S, MYC-HIS-RIPK1, GFP-RIPK3, HA-Ub, HA-K11 Ub, HA-K33 Ub, HA-K48 Ub, HA-K63 Ub, and FLAG-Vector were constructed by GeneChem (Shanghai, China). All plasmid constructs were validated by DNA sequencing. The plasmids were transfected into NIH/3T3 cells with Lipofectamine 3000 (Thermo Fisher Scientific, L3000015) according to the manufacturer's protocols. Normally, cells in a 6 cm plate were transfected with 1000 ng of plasmids. For transfection experiments with increasing amounts of plasmids, 500 ng or 1000 ng of plasmids were transfected into cells in a 6 cm plate.

## RNA isolation and qRT-PCR

TRIzol Reagent (Thermo Fisher Scientific, 15596026CN) was used to isolate total RNA from cells or tissues. The isolated RNA was then reverse-transcribed into cDNA with the PrimeScript™ RT reagent Kit (Takara, RR037A). After that, qRT-PCR was performed on a QuantStudio™ 5 Real-Time PCR System (Thermo Fisher Scientific) using specific primers and TB Green® Premix Ex Taq™ II (Takara, RR820A). All primers were synthesized by Sangon Biotech and the primer sequences are shown in Appendix Table S1.

## TTC staining

After euthanasia, the mice were intracardially perfused with PBS. The brains were isolated and snap frozen at −80 °C for 1 min. Then, each brain was cut into 6 slices with a brain slicing mold (RWD, 68707). The slices were stained with 2% TTC (Sigma-Aldrich, T8877) for 15 min at 37 °C, and then fixed in 4% paraformaldehyde (Solarbio, P1110). The infarct area of each slice was calculated using the ImageJ software (NIH Image). Infarct size (%) = (contralateral area − ipsilateral non-infarct area)/contralateral area × 100%.

## Neurological function assessment

The modified neurological severity score (mNSS), rotarod test, corner turning test, and adhesive removal test were applied to evaluate neurological function. Prior to MCAO operation, the mice were trained for these tests for three consecutive days. The assessment was performed by an examiner who was blinded to the group allocation of tested mice.

## Measurement of vascular physiology

Under anesthesia with isoflurane, cerebral blood flow and partial pressure of oxygen in $Otub2^{+/+}$ and $Otub2^{-/-}$ mice was measured by a laser Doppler monitor. The systolic blood pressure of $Otub2^{+/+}$ and $Otub2^{-/-}$ mice was measured by a blood pressure monitor (Softron Biotechnology). Each mouse was monitored for 5 times and the average value was calculated and displayed.

## OGD treatment

Cells cultured in glucose-free DMEM (Gibco, 11966025) were placed in a hypoxic chamber (Billups-Rothenberg) filled with 95% $N_2$ and 5% $CO_2$, followed by incubation at 37 °C for 6 h. After that, the cells were removed from the hypoxic chamber and the medium was changed back to standard culture medium. Subsequently, the cells were further cultured in a standard cell incubator containing 95% air and 5% $CO_2$ for 12 h.

## Induction and measurement of cell death

Cells were treated with 20 ng/ml of TNF-α (PeproTech, 315-01A) and 50 μM of CHX (Merck, 239763-M) to induce apoptosis. For the induction of necroptosis, cells were pre-treated with 50 μM of CHX and 20 μM of Z-VAD-FMK (MedChemExpress, HY-16658B) for 0.5 h, and then 20 ng/ml of TNF-α was added to induce cell death. Cell viability was determined by CCK-8 (New Cell & Molecular Biotech, C6005) according to the producer's instructions. The absorbance at 450 nm was measured with a Multiskan SkyHigh Microplate Spectrophotometer (Thermo Fisher Scientific).

## Administration of inhibitors

In certain experiments, cells were treated with 50 μM of Nec-1 (MedChemExpress, HY-15760), 25 μM of GSK-872 (MedChemExpress, HY-101872), 1 μM of GW806742X (MedChemExpress, HY-112292), or 20 μM of LN5P45 (MedChemExpress, HY-149482), which were all diluted in 1% DMSO. For in vivo treatment, mice were intraperitoneally injected with GSK-872 or LN5P45, both of which were diluted in corn oil (MedChemExpress, HY-Y1888). The control groups received an equivalent amount of vehicle solvent.

## Western blot analysis

Total protein was extracted with protein extraction reagent (Boster Bio, AR0101/AR0103) supplemented with Protease Inhibitor Cocktail (APE×BIO, K4002) and Phosphatase Inhibitor Cocktail (APE×BIO, K1015). After 30 min of incubation on ice, the lysates were centrifuged at 12,000 rpm for 15 min at 4 °C. The supernatant was collected and protein concentration was determined with the

Quick Start™ Bradford Protein Assay Kit (BIO-RAD, 5000201). Thereafter, the protein samples were mixed with 5× DualColor Protein Loading Buffer (FUDE Biological Technology, FD006) and heated at 100 °C for 10 min. Proteins were separated by SDS-PAGE and subsequently transferred to PVDF membranes (Cytiva, 10600023). After blocking with 5% skim milk (Biofroxx, 1172GR500), the membranes were incubated with primary antibodies overnight at 4 °C. On the next day, the membranes were incubated with corresponding secondary antibodies for 1 h at room temperature. Finally, the membranes were developed with Ncm ECL Ultra (New Cell & Molecular Biotech, P10300). The Fusion FX.EDGE system (Vilber) was used to capture Western blot images and the ImageJ software was applied to quantify band intensity.

## Immunoprecipitation

Protein samples were incubated with BeyoMag™ Protein A + G Magnetic Beads (Beyotime, P2108) for 2 h at 4 °C with gentle rotation to remove non-specific binding proteins. The beads were removed by centrifugation, and the samples were then incubated with antibodies overnight at 4 °C with gentle rotation. Thereafter, BeyoMag™ Protein A + G Magnetic Beads were added and incubated for 4 h at 4 °C with gentle rotation to capture immunocomplexes. The beads were collected by centrifugation and washed 5 times with ice-cold PBS before further analysis.

## In vitro deubiquitination assay

FLAG-OTUB2 and GFP-RIPK3 + HA-K48 Ub plasmids were transfected into different dishes of NIH/3T3 cells for 24 h, and the cells were then lysed for protein extraction. Immunoprecipitation was performed with anti-FLAG and anti-GFP antibodies to harvest OTUB2 and K48 polyubiquitinated RIPK3, respectively. After thorough wash with PBS, the beads were rinsed with deubiquitination buffer (5 mM $MgCl_2$, 2 mM ATP, 2 mM DTT, 50 mM Tris-HCl, 5% glycerol). Thereafter, K48 polyubiquitinated RIPK3 was incubated in the deubiquitination buffer at 37 °C for 2 h in the presence or absence of OTUB2.

## Histological analysis

Mice were sacrificed and then perfused sequentially with PBS and 4% paraformaldehyde. Brain tissue was fixed in 4% paraformaldehyde for 24 h. After fixation, the tissue was dehydrated and then embedded in paraffin. Subsequently, the paraffin-embedded tissue was cut into 20 μm slices. The slices were stained with Nissl Staining Solution (Beyotime, C0117) and H&E Staining Solution (Solarbio, G1120) according to the manufacturers' instructions.

## Immunofluorescence

Brain tissue was cut into 5 μm (regular imaging) or 20 μm (three-dimentional imaging) slices. Cultured cells were fixed in 4% paraformaldehyde for 15 min. The samples were then permeabilized with 0.5% Triton X-100 (Solarbio, T8200) for 30 min. After permeabilization, the samples were blocked with 5% BSA Blocking Buffer (Solarbio, SW3015) for 60 min, followed by incubation with primary antibodies overnight at 4 °C. On the next day, after incubation with fluorescence-conjugated secondary antibodies (Yeasen

Biotechnology) for 1 h at 37 °C, the samples were subjected to DAPI staining (Solarbio, S2110). Certain samples were stained with the TUNEL staining kit (Proteintech, PF00006) for the detection of apoptosis. Fluorescence images were taken on a FV3000 Confocal Laser Scanning Microscope (Olympus) and analyzed by the ImageJ software.

## LC-MS/MS analysis

One hour after MCAO surgery, 20 mg/kg of GSK-872 or LN5P45 was intraperitoneally injected into each mouse. After 1 or 3 h, the brain tissue was isolated and homogenized in sterile PBS, followed by treatment with acetonitrile and methanol. After sonication, the samples were centrifuged at 12,000 rpm for 15 min at 4 °C. The supernatant was treated again with acetonitrile and subsequently filtered with 0.22 μm filters. GSK-872 or LN5P45 in the samples were detected by the liquid chromatography-triple quadrupole mass spectrometer (SCIEX). All data were analyzed by the MultiQuant™ 3.0.3 software (SCIEX).

## Generation of human brain organoids

Human brain organoids were differentiated from H9 hESCs with the STEMdiff™ Cerebral Organoid Kit (STEMCELL Technologies, 08570). Briefly, H9 hESCs (9000 cells/well) were inoculated in each well of the round-bottom ultra-low attachment 96-well plate (Corning, 7007) and cultured in the Formation Medium containing the selective ROCK inhibitor Y-27632 (STEMCELL Technologies, 72302). On day 5, the embryoid bodies were transferred to a 24-well ultra-low attachment plate (Corning, 3473) containing the Induction Medium. The Induction Medium contained STEMdiff™ Cerebral Organoid Basal Medium 1 (STEMCELL Technologies, 08572) and STEMdiff™ Cerebral Organoid Basal Supplement B (STEMCELL Technologies, 08575). After two days of culture, the embryoid bodies were embedded in the Matrigel and then transferred to a 6-well ultra-low adherent plate (12–16 embryoid bodies/well). On day 10, the organoids were cultured in the Maturation Medium on an orbital shaker (60–90 RPM) in a cell incubator. The Maturation Medium contained STEMdiff™ Cerebral Organoid Basal Medium 2 (STEMCELL Technologies, 08573) and STEMdiff™ Cerebral Organoid Basal Supplement E (STEMCELL Technologies, 08578). The medium was changed every 3–4 days until day 30. Mature organoids were subjected to OGD treatment and subsequently analyzed by immunofluorescence.

## Statistical analysis

Statistical analyses were carried out using the GraphPad Prism 8 software (GraphPad). The two-tailed unpaired Student's t test and the Mann–Whitney U test were conducted to compare the two sets of data that were in normal distribution and those that did not conform to normal distribution, respectively. Multiple Student's t tests were applied for comparisons between two groups under multiple conditions. The two-way ANOVA test was utilized for multiple comparisons among three or more groups. All experiments were repeated at least twice, and the data are expressed as mean ± SEM. $P$ values < 0.05 were considered significant. The number of samples for each experiment is shown in figures or legends. Mice of the same genotype were randomly allocated to experimental groups. Blinding was not applied in experiments except for the neurological function assessment, which was performed in a blinded manner. Approximately 10% mice died during the MCAO surgery and

### The paper explained

**Problem**

Ischemic stroke is a leading cause of mortality and disability around the world. Neurons rapidly die after cerebral ischemia, causing severe and irreversible damage. However, the mechanisms regulating neuronal death after ischemic stroke remain incompletely understood. Identification of proteins that critically regulate neuronal death may provide novel therapeutic targets for ischemic stroke.

**Results**

In this study, we demonstrate that cerebral ischemia-induced neuronal death is critically regulated by the deubiquitinating enzyme Otubain-2 (OTUB2) and deletion of OTUB2 ameliorates ischemic stroke injury in mice. OTUB2 enhances neuronal necroptosis, a predominant form of cell death after ischemic stroke, by maintaining the protein levels of receptor-interacting protein kinase 3 (RIPK3). Consistently, inhibition of OTUB2 with the small-molecule inhibitor LN5P45 attenuates ischemic stroke injury in mice and reduces neuronal death in human brain organoids after oxygen-glucose deprivation.

**Impact**

Our results reveal that genetic deletion or pharmacological inhibition of OTUB2 ameliorates ischemic stroke injury, validating OTUB2 as a druggable target for the treatment of ischemic stroke.

these mice were excluded from further analysis. Detailed statistical information is shown in Appendix Tables S2–17.

## Data availability

This study includes no data deposited in external repositories.

The source data of this paper are collected in the following database record: biostudies:S-SCDT-10_1038-S44321-025-00206-6.

## Peer review information

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

## Acknowledgements

This work was supported by grants from the Natural Science Foundation of Zhejiang Province (LZ24H090003) and National Natural Science Foundation of China (81900496) to XW.

## Author contributions

**Fuqi Mei**: Conceptualization; Resources; Data curation; Formal analysis; Investigation; Visualization; Methodology; Writing—original draft; Writing—review and editing. **Deyu Deng**: Data curation; Investigation; Visualization; Methodology; Writing—review and editing. **Zijun Cao**: Data curation; Investigation; Visualization; Methodology. **Liyan Lou**: Investigation; Methodology. **Kangmin Chen**: Investigation; Methodology. **Minjie Hu**: Investigation; Methodology. **Zhenhu Zhu**: Resources; Investigation. **Jiangyun Shen**: Resources; Investigation. **Jianzhao Zhang**: Resources; Investigation. **Jie Liang**: Resources; Investigation. **Jingyong Huang**: Resources; Writing—review and editing. **Min Bao**: Validation; Writing—review and editing. **Ari Waisman**: Writing—review and editing. **Xu Wang**: Conceptualization; Resources; Supervision; Funding acquisition; Validation; Writing—original draft; Project administration; Writing—review and editing.

Source data underlying figure panels in this paper may have individual authorship assigned. Where available, figure panel/source data authorship is listed in the following database record: biostudies:S-SCDT-10_1038-S44321-025-00206-6.

## Disclosure and competing interests statement

The authors declare no competing interests.

