## [Peer Review File · EMBO Molecular Medicine]

Deubiquitination of RIPK3 by OTUB2 potentiates neuronal necroptosis after ischemic stroke

Fuqi Mei, Deyu Deng, Zijun Cao, Liyan Lou, Kangmin Chen, Minjie Hu, Zhenhu Zhu, Jiangyun Shen, Jianzhao Zhang, Jie Liang, Jingyong Huang, Min Bao, Ari Waisman, and Xu Wang

Corresponding author: Xu Wang (wangxu@ojlab.ac.cn)

Review Timeline:

Submission Date:	23rd Aug 24
Editorial Decision:	25th Sep 24
Revision Received:	19th Jan 25
Editorial Decision:	13th Feb 25
Revision Received:	14th Feb 25
Accepted:	19th Feb 25

Editor: Zeljko Durdevic

Transaction Report:

25th Sep 2024

Dear Prof. Wang,

Thank you for the submission of your manuscript to EMBO Molecular Medicine. We have now received feedback from the three reviewers who agreed to evaluate your manuscript. As you will see from the reports pasted below, all three referees recognize interest of the study but also raise important concerns that should be addressed in a major revision. Particular attention should be given to better emphasizing the novelty of the study, providing more detailed information about statistics, providing evidence that the compound GSK-872 crosses blood-brain barrier, performing dose escalation experiment and repeating treatment of the mouse model in more clinically relevant setting (e.g. at least 1-3 hours after occlusion). Given that the revision will require extensive experimentation we think six months rather than three months would be more appropriate to provide the complete revision. If you would like to discuss further the points raised by the referees, I am available to do so via email or video. Let me know if you are interested in this option.

We would welcome the submission of a revised version within six months for further consideration. Please let us know if you require longer to complete the revision.

I look forward to receiving your revised manuscript.

Yours sincerely,

Zeljko Durdevic

We require:

- 1) A .docx formatted version of the manuscript text (including legends for main figures, EV figures and tables). Please make sure that the changes are highlighted to be clearly visible.
- 2) Individual production quality figure files as .eps, .tif, .jpg (one file per figure). For guidance, download the 'Figure Guide PDF': (<https://www.embopress.org/page/journal/17574684/authorguide#figureformat>).
- 3) A .docx formatted letter INCLUDING the reviewers' reports and your detailed point-by-point responses to their comments. As part of the EMBO Press transparent editorial process, the point-by-point response is part of the Review Process File (RPF), which will be published alongside your paper.
- 4) A complete author checklist, which you can download from our author guidelines (<https://www.embopress.org/page/journal/17574684/authorguide#submissionofrevisions>). Please insert information in the checklist that is also reflected in the manuscript. The completed author checklist will also be part of the RPF.

6) It is mandatory to include a 'Data Availability' section after the Materials and Methods. Before submitting your revision, primary datasets produced in this study need to be deposited in an appropriate public database, and the accession numbers and database listed under 'Data Availability'. Please remember to provide a reviewer password if the datasets are not yet public (see <https://www.embopress.org/page/journal/17574684/authorguide#dataavailability>).

12) Author contributions: You will be asked to provide CRediT (Contributor Role Taxonomy) terms in the submission system. These replace a narrative author contribution section in the manuscript.

13) A Conflict of Interest statement should be provided in the main text.

14) Every published paper now includes a 'Synopsis' to further enhance discoverability. Synopses are displayed on the journal webpage and are freely accessible to all readers. They include a short stand first (maximum of 300 characters, including space) as well as 2-5 one-sentences bullet points that summarizes the paper. Please write the bullet points to summarize the key NEW findings. They should be designed to be complementary to the abstract - i.e. not repeat the same text. We encourage inclusion of key acronyms and quantitative information (maximum of 30 words / bullet point). Please use the passive voice. Please attach these in a separate file or send them by email, we will incorporate them accordingly.

15) Include a Reagents and Tools Table as part of the Methods section, which can be downloaded from our author guidelines (<https://www.embopress.org/page/journal/17574684/authorguide#structuredmethods>)

***** Reviewer's comments *****

Referee #1 (Comments on Novelty/Model System for Author):

- manuscript seems methodologically high quality
- Novelty needs clarification
- Medical impact needs to be improved and needs clarification
- methods and models are adequate and high quality.

Referee #1 (Remarks for Author):

Manuscript entitled "Deubiquitination of RIPK3 by OTUB2 potentiates neuronal necroptosis after ischemic stroke" by Mei et al., explores the role OTUB2 and RIPK3 proteins in regulating brain injury and outcome after ischemic stroke. The conclusion authors make is that OTUB2 critically regulates ischemic stroke injury by potentiating neuronal necroptosis and that OTUB2 inhibition could be a novel drug target for treating ischemic stroke.

Advanced methods were carried out e.g. hypoxia experiments in human organoids and methodologically the study seem to be sound.

Major concerns.

1. Previous study by the same group has shown that ischemic cerebral injury in mice is controlled by deubiquitinating enzymes OTUB2 and genetic ablation of OTUB2 ameliorated cerebral injury after ischemic stroke. Functionally, OTUB2 has increased neuronal death after ischemic insult by enhancing RIPK3-mediated necroptosis. Mechanistically, OTUB2 inhibited the degradation of RIPK3 through removing K48-linked polyubiquitin chains from RIPK3, thereby enhancing RIPK3 abundance and activation. Furthermore, pharmacological inhibition of OTUB2 attenuated ischemic stroke injury in mice and neuronal death in human brain organoids. The authors also indicate that their previous results provide a novel regulatory mechanism for necroptosis and identify a potential therapeutic target for ischemic stroke. Therefore, the current study, seems to be a follow up study of the previously and recently published work by the same group. It remains unclear to the reader what is the difference and novelty of this study as compared to the previous one.
2. Stroke studies are recommended to be done in both male and female mice and animals with different ages, and particularly in aged mice since biggest risk factor for stroke is age. The authors do not state the age of male mice used.
3. The study do not have any indications of randomizations, blinding and exclusion criteria.
4. No mention of power calculations nor analysis of normality in case of parametric tests used.
5. The study should greatly be improved by implementing statistical table indicating the test used for each panel in each figure, with exact p value and result of the test.
6. N values are rather small e.g. 3-4 in many figures, and group sizes seem not be to equal e.g. 3-7.
7. NeuN staining seems to be cytosolic. The staining pattern looks rather strange considering that NeuN should be transcription factor, Figure 2A.
8. What happen if OTUB2 would remove K48 ubiquitination from RIPK3 after stroke has happened e.g. during the recovery days 3-7 post stroke.
9. The neuroprotective is not really seen feasible in terms of translational feasibility for stroke.
10. Mechanistic studies employing also overexpression experiments and downstream signalling modulators are missing.
11. Does OTUB2 deletion alter blood gas, blood pressure, cerebral blood flow?

Minor comments.

- The manuscripts contain some spelling mistakes e.g. CRISPER

Referee #2 (Remarks for Author):

In the present work work, the authors found that the deletion of the deubiquitinating enzyme OTUB2, an enzyme which is expressed in neurons, decreased the expression of the receptor-interacting protein kinase 3 (RIPK3), which is involved in mediated-neuronal necroptosis. OTUB2 deletion caused also a reduction of the ischemic brain volume induced by transient middle cerebral artery occlusion (tMCAO). The authors found that OTUB2 interacts with RIPK3 protein, inhibiting its proteasomal degradation and therefore triggering necroptosis. In addition, they found that the pharmacological inhibition of OTUB2 with the compound GSK-872, a putative RIPK3 inhibitor, alleviated ischemic brain injury in mice and reduced ischemia-induced neuronal death in human brain organoids

MAJOR POINTS

The experimental design of the present work is well structured, with adequate methodologies to pursue the hypothesized scientific objectives.

However, there are some scientific aspects that require some additional experiments :

1. The experiment with pharmacological inhibition of OTUB2 with the putative RIPK3 inhibitor GSK-872 needs at least two additional increasing doses. The single in vivo dose of GSK-872 is not sufficient to demonstrate the relationship between RIPK3 inhibition and the reduction of the volume of the ischemic core. These additional experiments are necessary especially if one considers that the reduction of the ischemic volume in OTUB2 ^{-/-} mice, although it is statistically significant, is of small entity. Furthermore, the pharmacology requires a dose-effect of a compound to establish a relationship between drug administration and the observed effects.
2. No evidence was provided in the paper on the possibility that GSK- 872 crosses the blood-brain barrier reaching the brain, where it should exert its inhibitory action on cerebral RIPK3.
3. The authors propose GSK, or generally a drug that inhibits RIPK3 activity, might be a new candidate for reducing the brain injury in the stroke, however to sustain this hypothesis in the animal model tMCAO used to reproduce the human pathology requires that the first administration of GSK-872 can not be performed at time Zero when the occlusion of the middle cerebral occlusion is performed. In fact this situation does not occur in humans, who usually receive the pharmacological intervention in the later hours from the insult. It would be more rationale to administer to rodents the candidate drug at least 1-3 hours after the occlusion.

MINOR POINTS

1. line 163/164 "was not aware of the group identify ? of tested mice".
2. line 374 ".in In line"
3. The concentration of the plasmids used for the transfection should be indicated.
4. In the antibody section, all the concentrations that were used for the experiments should be indicated
5. For the reproducibility of the work, the section of brain organoids should be expanded; indicate the composition of the induction medium and of the maturation medium. Indicate the agents used to detach the organoids from the plates. Indicate the markers that have been used for the characterization of the organoids, (i.e. markers of neural progenitors proliferation, neuronal differentiation, inhibitory and excitatory synapses).
6. Figure 1A, since the GAPDH control gene is highly modulated, I would recommend to repeat the Western blot.
7. Figure 3P, please repeat the p-RIPK Western blot. The kDa of each band for all the Western blot images should be indicated.
8. For most of the co-IP figures, an arrow and the kDa of the band of interest should be indicated.
9. Line 447: write in the text on which kind of cell LN5P45 treatment was tested for the dose assay.
10. In Supplementary figure 10 the Otub2^{-/-} bar is missing.
11. Why do the authors use NIH-3T3 cells as an in vitro model? Was the differentiation induced? This point should be clarified.

Referee #3 (Remarks for Author):

The manuscript by Mei et al showed that RIPK3 can be deubiquitinated by OTUB2 and demonstrated that the deubiquitination of RIPK3 by OTUB2 plays a role in neuronal necroptosis in ischemic stroke. This is the first report on the role of OTUB2 in necroptosis and in the ischemia model. The authors also showed that OTUB2 removes K48-linked polyubiquitin chains from RIPK3 and thus reduces proteasomal degradation of RIPK3, and that OTUB2 inhibition could be a therapeutic approach for treating ischemic stroke. Overall, the study was well conducted and the conclusion was supported by the data. However, there are some issues needed to be addressed.

1. Based on the data presented in the manuscript, OTUB2 functions to control RIPK3 level in the steady state, in other words, it is stimulation-independent. The authors need to clarify it.
2. If the above comment stands, the statements such as "OTUB2 deficiency significantly reduced the phosphorylation of RIPK3 and MLKL in primary neurons after OGD" in the text shall be revised.
3. Background information of OTUB2 should be provided in the Introduction.
4. MACO, OTUB2, OGD, etc., should be spelled out when they first appear in the Introduction and Results. The reason why HT22 cell was used should be explained when it first appears in the Results.
5. Reduction of K48 ubiquitination of RIPK3 by OTUB2 was shown in Figure 5. But the authors also need to show whether OTUB2 had effects on other types of ubiquitinations on RIPK3.

***** Reviewer's comments *****

Referee #1 (Comments on Novelty/Model System for Author):

- manuscript seems methodologically high quality
- Novelty needs clarification
- Medical impact needs to be improved and needs clarification
- methods and models are adequate and high quality.

Referee #1 (Remarks for Author):

Manuscript entitled "Deubiquitination of RIPK3 by OTUB2 potentiates neuronal necroptosis after ischemic stroke" by Mei et al., explores the role OTUB2 and RIPK3 proteins in regulating brain injury and outcome after ischemic stroke. The conclusion authors make is that OTUB2 critically regulates ischemic stroke injury by potentiating neuronal necroptosis and that OTUB2 inhibition could be a novel drug target for treating ischemic stroke.

Advanced methods were carried out e.g. hypoxia experiments in human organoids and methodologically the study seem to be sound.

Major concerns.

1. Previous study by the same group has shown that ischemic cerebral injury in mice is controlled by deubiquitinating enzymes OTUB2 and genetic ablation of OTUB2 ameliorated cerebral injury after ischemic stroke. Functionally, OTUB2 has increased neuronal death after ischemic insult by enhancing RIPK3-mediated necroptosis. Mechanistically, OTUB2 inhibited the degradation of RIPK3 through removing K48-linked polyubiquitin chains from RIPK3, thereby enhancing RIPK3 abundance and activation. Furthermore, pharmacological inhibition of OTUB2 attenuated ischemic stroke injury in mice and neuronal death in human brain organoids. The authors also indicate that their previous results provide a novel regulatory mechanism for necroptosis and identify a potential therapeutic target for ischemic stroke. Therefore, the current study, seems to be a follow up study of the previously and recently published work by the same group. It remains unclear to the reader what is the difference and novelty of this study as compared to the previous one.

Response: We are sorry for the confusions. Actually, we have never published any papers about OTUB2 in ischemic cerebral injury. Previously, we have published a paper showing that another deubiquitinating enzyme, USP25, reduces ischemic stroke injury by inhibiting microglia-mediated neuroinflammation (Li et al. *Adv Sci.* 2023). We have also shown that USP25 contributes to Alzheimer's disease (Zheng et al. *J Clin Invest.* 2022). However, the functional roles and molecular mechanisms of USP25 and OTUB2 are totally different. In the present study, we demonstrate that OTUB2 potentiates ischemia-induced neuronal necroptosis by stabilizing RIPK3, a pivotal molecule inducing necroptosis (Zhang et al. *Science.* 2009; He et al. *Cell.* 2009). Although multiple E3s have been shown to ubiquitinate RIPK3 for degradation (Seo et al. *Nat Cell Biol.* 2016; Choi et al. *Mol Cell.* 2018; Xie et al. *J Clin Invest.* 2020; Mei et al. *Cell Death Differ.* 2021), as far as we know, OTUB2 is the only enzyme known to stabilize RIPK3, revealing an important regulatory mechanism for RIPK3. In this study, we show that the genetic ablation or

pharmacological inhibition of OTUB2 with the small-molecule inhibitor LN5P45 ameliorates ischemic stroke injury in mice. Importantly, OTUB2 inhibition by LN5P45 inhibited OGD-induced neuronal death in human brain organoids, validating the potential of OTUB2 as a drug target for ischemic stroke. Therefore, we think our findings in this study are novel and important, and this study is not a direct follow-up study of any published work. We have specified the novelties and importance of this study in the Introduction, Discussion, and Synopsis of this manuscript.

2. Stroke studies are recommended to be done in both male and female mice and animals with different ages, and particularly in aged mice since biggest risk factor for stroke is age.

The authors do not state the age of male mice used.

Response: We agree with the reviewer that occurrence of stroke is strongly influenced by gender and age. However, in this study, we aimed to study the functional role of OTUB2 in ischemic stroke-induced neuronal death and cerebral injury. For the study of ischemic stroke injury, male mice are commonly used (Bertrand et al. Nat Commun. 2019; Li et al. EMBO Mol Med. 2020; Lemmerman et al. Sci Adv. 2021; Liu et al. Stroke. 2023).

As suggested by the reviewer, we have performed new MCAO experiments with aged male mice. In line with our previous results obtained with young adult mice, OTUB2 deficiency significantly reduced MCAO-induced cerebral infarction and neurological deficits in aged mice (Appendix Fig. S1A-B, S2A-C). In addition, we have specified the age of male mice used in this study, lines 392-395.

3. The study do not have any indications of randomizations, blinding and exclusion criteria.

Response: As requested by the reviewer, we have added the missing information in the Materials and Methods section, “Mice of the same genotype were randomly allocated to experimental groups. Blinding was not applied in experiments except for the neurological function assessment, which was performed in a blinded manner. Approximately 10% mice died during the MCAO surgery and these mice were excluded from further analysis”, lines 568-573.

4. No mention of power calculations nor analysis of normality in case of parametric tests used.

Response: As suggested by the reviewer in comments 4 and 5, we have added statistical tables for each panel in each figure, specifying analysis of normality, statistical methods, power, and exact p

values (Appendix Tables S2-S17).

5. The study should greatly be improved by implementing statistical table indicating the test used for each panel in each figure, with exact p value and result of the test.

Response: As mentioned above, the statistical tables have been added (Appendix Tables S2-S17).

6. N values are rather small e.g. 3-4 in many figures, and group sizes seem not be to equal e.g. 3-7.

Response: For results with small group sizes, we have included more biological replicates to improve the credibility (Figures 1B, 3N, 4B-C, Appendix Figures S6B-C, S6F-I).

The group sizes in Figures 2I-M were not equal (3-7) because 3 mice were used for control groups and 7 mice were used for MCAO groups. As suggested by the reviewer, we have added new biological replicates for the control group (Figures 2I-M).

7. NeuN staining seems to be cytosolic. The staining pattern looks rather strange considering that NeuN should be transcription factor, Figure 2A.

Response: We thank the reviewer for the careful review of our manuscript. We agree with the reviewer that the old staining was not appropriate. We have performed new experiments and the results are included in the new Figure 2A.

8. What happen if OTUB2 would remove K48 ubiquitination from RIPK3 after stroke has happened e.g. during the recovery days 3-7 post stroke.

Response: To test this, we performed new experiments and found that OTUB2 deficiency increased the K48 ubiquitination of RIPK3 in the brain at day 3 after MCAO (Figure 5G). The results are similar to that in control mice (Figure 5F), showing that OTUB2 constitutively K48 deubiquitinates RIPK3. Consistently, OTUB2 deficiency is accompanied with lower levels of RIPK3 under resting conditions (Figures 3A, 3F, 3P, 4A-B, 4J, 5A-B).

9. The neuroprotective is not really seen feasible in terms of translational feasibility for stroke.

Response: We agree with the reviewer that it is still challenging to treat stroke, as multiple factors, such as neuronal death, oxidative stress, and inflammation, contribute to stroke injury. Of note, neuronal necroptosis is a core event after ischemic stroke. Shortly after ischemic stroke, neurons undergo necroptosis, followed by apoptosis (Naito et al. Proc Natl Acad Sci U S A. 2020). Blocking necroptosis with the RIPK1 inhibitor necrostatin-1 efficiently reduces ischemic stroke injury in mice (Degterev et al. Nat Chem Biol. 2005). In this study, we show that OTUB2 regulates necroptosis by stabilizing RIPK3, an indispensable signaling molecule in the necroptotic pathway (Zhang et al. Science. 2009; He et al. Cell. 2009). In mice, genetic deletion or pharmacological inhibition of OTUB2 significantly attenuated ischemic stroke injury, indicating OTUB2 inhibition may be a potential strategy for treating stroke. Honestly, we admit that there are huge differences between human stroke and the MCAO mouse model and uncertainties exist for the clinical translation of our findings. Like most drugs, numerous pre-clinical and clinical tests will be required for the translation of our findings. Despite these problems, our study with mice and human brain organoids unequivocally elucidates the role of OTUB2 in ischemia-induced neuronal death and thereby provides a potential therapeutic approach for stroke.

10. Mechanistic studies employing also overexpression experiments and downstream signalling modulators are missing.

Response: As suggested by the reviewer, we have performed corresponding experiments. Overexpression of OTUB2, rather than the C51S inactive mutant, significantly increased RIPK3 protein levels and TCZ-induced necroptosis, showing that OTUB2 regulates necroptosis by deubiquitinating RIPK3 through its catalytic activity (Figures 5J-K, 6H). In the necroptotic pathway, active RIPK3 phosphorylates the downstream signaling molecule MLKL, which subsequently causes membrane rupture (Sun et al. Cell. 2012; Wang et al. Mol Cell. 2014). We found that inhibition of MLKL efficiently blocked TCZ-induced necroptosis and blunted the difference between *Otub2*^{+/+} and *Otub2*^{-/-} cells, suggesting that OTUB2 regulates necroptosis (Figure 6G).

11. Does OTUB2 deletion alter blood gas, blood pressure, cerebral blood flow?

Response: As suggested by the reviewer, we have performed new experiments to measure these parameters. The results indicate that OTUB2 deletion does not alter partial pressure of oxygen, blood pressure, and cerebral blood flow (Appendix Figure S1E-I).

Minor comments.

- The manuscripts contain some spelling mistakes e.g. CRISPER

Response: We thank the reviewer for pointing out this mistake. We have corrected it in the

manuscript, line 369.

References:

- Choi SW et al. PELI1 Selectively Targets Kinase-Active RIP3 for Ubiquitylation-Dependent Proteasomal Degradation. *Mol Cell*. 2018;70(5):920-935.e7.
- Mei P et al. E3 ligase TRIM25 ubiquitinates RIP3 to inhibit TNF induced cell necrosis. *Cell Death Differ*. 2021;28(10):2888-2899.
- Seo J et al. CHIP controls necroptosis through ubiquitylation- and lysosome-dependent degradation of RIPK3. *Nat Cell Biol*. 2016;18(3):291-302.
- Xie Y et al. Gut epithelial TSC1/mTOR controls RIPK3-dependent necroptosis in intestinal inflammation and cancer. *J Clin Invest*. 2020;130(4):2111-2128.
- Li Z et al. USP25 Inhibits Neuroinflammatory Responses After Cerebral Ischemic Stroke by Deubiquitinating TAB2. *Adv Sci (Weinh)*. 2023;10(28):e2301641.
- Zheng Q et al. USP25 inhibition ameliorates Alzheimer's pathology through the regulation of APP processing and A β generation. *J Clin Invest*. 2022;132(5):e152170.
- He S et al. Receptor interacting protein kinase-3 determines cellular necrotic response to TNF-alpha. *Cell*. 2009;137(6):1100-1111.
- Zhang DW et al. RIP3, an energy metabolism regulator that switches TNF-induced cell death from apoptosis to necrosis. *Science*. 2009;325(5938):332-336.
- Liu X et al. BAG3 Overexpression Attenuates Ischemic Stroke Injury by Activating Autophagy and Inhibiting Apoptosis. *Stroke*. 2023;54(8):2114-2125.
- Li Q et al. Inhibition of double-strand DNA-sensing cGAS ameliorates brain injury after ischemic stroke. *EMBO Mol Med*. 2020;12(4):e11002.
- Bertrand L et al. Targeting the HIV-infected brain to improve ischemic stroke outcome. *Nat Commun*. 2019;10(1):2009.
- Lemmerman LR et al. Nanotransfection-based vasculogenic cell reprogramming drives functional recovery in a mouse model of ischemic stroke. *Sci Adv*. 2021;7(12):eabd4735.
- Naito MG et al. Sequential activation of necroptosis and apoptosis cooperates to mediate vascular and neural pathology in stroke. *Proc Natl Acad Sci U S A*. 2020;117(9):4959-4970.
- Degterev A et al. Chemical inhibitor of nonapoptotic cell death with therapeutic potential for ischemic brain injury. *Nat Chem Biol*. 2005;1(2):112-119.
- Sun L et al. Mixed lineage kinase domain-like protein mediates necrosis signaling downstream of RIP3 kinase. *Cell*. 2012;148(1-2):213-227.
- Wang H et al. Mixed lineage kinase domain-like protein MLKL causes necrotic membrane disruption upon phosphorylation by RIP3. *Mol Cell*. 2014;54(1):133-146.

Referee #2 (Remarks for Author):

In the present work work, the authors found that the deletion of the deubiquitinating enzyme OTUB2, an enzyme which is expressed in neurons, decreased the expression of the receptor-interacting protein kinase 3 (RIPK3), which is involved in mediated-neuronal necroptosis. OTUB2 deletion caused also a reduction of the ischemic brain volume induced by transient middle cerebral artery occlusion (tMCAO).The authors found that OTUB2 interacts with RIPK3 protein, inhibiting its proteasomal degradation and therefore triggering necroptosis. In addition, they found that the pharmacological inhibition of OTUB2 with the compound GSK-872, a putative RIPK3 inhibitor, alleviated ischemic brain injury in mice and reduced ischemia-induced neuronal death in human brain organoids

MAJOR POINTS

The experimental design of the present work is well structured, with adequate methodologies to pursue the hypothesized scientific objectives.

However, there are some scientific aspects that require some additional experiments :

1. The experiment with pharmacological inhibition of OTUB2 with the putative RIPK3 inhibitor GSK-872 needs at least two additional increasing doses. The single in vivo dose of GSK-872 is not sufficient to demonstrate the relationship between RIPK3 inhibition and the reduction of the volume of the ischemic core. These additional experiments are necessary especially if one considers that the reduction of the ischemic volume in OTUB2

-/- mice, although it is statistically significant, is of small entity. Furthermore, the pharmacology requires a dose-effect of a compound to establish a relationship between drug administration and the observed effects.

Response: As suggested by the reviewer, we have performed new experiments with 10 mg/kg and 20 mg/kg of GSK-872 (RIPK3 inhibitor) and LN5P45 (OTUB2 inhibitor). In addition, as suggested by the reviewer in comment 3, GSK-872 and LN5P45 were administered at 1 hour after MCAO. As shown in the new Figure 7A-I, GSK-872 and LN5P45 can significantly attenuate ischemic stroke injury in mice at the two indicated doses. Compared with the low dose (10 mg/kg), the high dose (20 mg/kg) of GSK-872 and LN5P45 slightly reduced ischemic stroke injury in *Otub2*^{+/+} mice. However, the low dose (10 mg/kg) is enough for GSK-872 and LN5P45 to efficiently attenuate ischemic stroke injury. The new results are consistent with our old results.

2. No evidence was provided in the paper on the possibility that GSK- 872 crosses the blood-brain barrier reaching the brain, where it should exert its inhibitory action on cerebral RIPK3.

Response: We thank the reviewer for pointing out this negligence. Previously, we predicted that GSK-872 and LN5P45 could cross the BBB using an online tool (www.graphbbb.com) before the animal experiments. Indeed, a direct experimental evidence confirming that the inhibitors cross the BBB is essential for this study. Therefore, we performed experiments and confirmed that both GSK-872 and LN5P45 could cross the BBB. The results are shown in Appendix Figures S12,S14.

3. The authors propose GSK, or generally a drug that inhibits RIPK3 activity, might be a new candidate for reducing the brain injury in the stroke, however to sustain this hypothesis in the animal model tMCAO used to reproduce the human pathology requires that the first administration of GSK-872 can not be performed at time Zero when the occlusion of the middle cerebral occlusion is performed. In fact this situation does not occur in humans, who usually receive the pharmacological intervention in the later hours from the insult. It would be more rationale to administer to rodents the candidate drug at least 1-3 hours after the occlusion.

Response: We agree with the reviewer that it makes more sense to give the pharmacological intervention after the MCAO surgery. As suggested by the reviewer, GSK-872 and LN5P45 were intraperitoneally injected at 1 hour after the MCAO surgery, and these drugs significantly reduced ischemic stroke injury in mice (Figure 7A-D, 7F-I).

MINOR POINTS

1. line 163/164 "was not aware of the group identify ? of tested mice".

Response: We have rephrased the sentence as “The assessment was performed by an examiner who was blinded to the group allocation of tested mice”, lines 437-438.

2. line 374 ".in In line"

Response: We thank the reviewer for the careful review of our manuscript and pointing out this mistake. The mistake has been corrected, line 190.

3. The concentration of the plasmids used for the transfection should be indicated.

Response: We have specified the amounts of plasmids used for the transfection, lines 410-413.

4. In the antibody section, all the concentrations that were used for the experiments should be indicated

Response: As requested by the reviewer, we have added the concentrations of antibodies to the Reagents and Tools Table, line 698.

5. For the reproducibility of the work, the section of brain organoids should be expanded; indicate the composition of the induction medium and of the maturation medium. Indicate the agents used to detach the organoids from the plates. Indicate the markers that have been used for the characterization of the organoids, (i.e. markers of neural progenitors

proliferation, neuronal differentiation, inhibitory and excitatory synapses).

Response: As suggested by the reviewer, we have expanded the generation of brain organoids. We have specified the composition of the induction medium and the maturation medium. The brain organoids were cultured in ultra-low adherent plates and they were loosely attached to the plates. Mature brain organoids were positive for NeuN (Figure 7K-M). The modified experimental procedures were included in the Methods section, lines 374-375, 540-557.

6. Figure 1A, since the GAPDH control gene is highly modulated, I would recommend to repeat the Western blot.

Response: We have replaced Figure 1A with results from repeated experiments.

7. Figure 3P, please repeat the p-RIPK Western blot. The kDa of each band for all the Western blot images should be indicated.

Response: We have replaced these blots with results from repeated experiments. In addition, we have added markers for all blots.

8. For most of the co-IP figures, an arrow and the kDa of the band of interest should be indicated.

Response: We have added markers for all blots including blots of the co-IP experiments.

9. Line 447: write in the text on which kind of cell LN5P45 treatment was tested for the dose assay.

Response: We have added HT-22 cells to the sentence, line 267-268.

10. In Supplementary figure 10 the *Otub2*^{-/-} bar is missing.

Response: We have added the missing *Otub2*^{-/-} bar.

11. Why do the authors use NIH-3T3 cells as an in vitro model? Was the differentiation

induced? This point should be clarified.

Response: NIH/3T3 cells are embryonic fibroblast cells and they are widely used for *in vitro* investigations. NIH/3T3 cells can be easily transfected with plasmids and are therefore used as “tool cells” for studying signal transduction (Zhao et al. Nat Commun. 2024; Shen et al. EMBO Rep. 2024). In this study, we used NIH/3T3 cells only for the transfection experiments. NIH/3T3 cells are different from 3T3-L1 cells, which can be differentiated into mature adipocytes.

References

Zhao Y et al. Podocyte OTUD5 alleviates diabetic kidney disease through deubiquitinating TAK1 and reducing podocyte inflammation and injury. Nat Commun. 2024;15(1):5441. Published 2024 Jun 27.

Shen J et al. YOD1 sustains NOD2-mediated protective signaling in colitis by stabilizing RIPK2. EMBO Rep. 2024;25(11):4827-4845.

Referee #3 (Remarks for Author):

The manuscript by Mei et al showed that RIPK3 can be deubiquitinated by OTUB2 and demonstrated that the deubiquitination of RIPK3 by OTUB2 plays a role in neuronal necroptosis in ischemic stroke. This is the first report on the role of OTUB2 in necroptosis and in the ischemia model. The authors also showed that OTUB2 removes K48-linked polyubiquitin chains from RIPK3 and thus reduces proteasomal degradation of RIPK3, and that OTUB2 inhibition could be a therapeutic approach for treating ischemic stroke. Overall, the study was well conducted and the conclusion was supported by the data. However, there are some issues needed to be addressed.

1. Based on the data presented in the manuscript, OTUB2 functions to control RIPK3 level in the steady state, in other words, it is stimulation-independent. The authors need to

clarify it.

Response: The reviewer is correct. We show in the manuscript that OTUB2 maintains the protein levels of RIPK3 in both resting and stimulated cells.

2. If the above comment stands, the statements such as "OTUB2 deficiency significantly reduced the phosphorylation of RIPK3 and MLKL in primary neurons after OGD" in the text shall be revised.

Response: In this study, we show that OTUB2 stabilizes RIPK3 by reducing its proteasomal degradation, thereby enhancing the activation of the RIPK3-MLKL necroptotic signaling. Indeed, this sentence causes misunderstanding that OTUB2 has a direct impact on the phosphorylation of RIPK3. Actually, the reduced phosphorylation is indirectly caused by reduced RIPK3 protein levels. We have rephrased the sentence to "OTUB2 deficiency significantly reduced the levels of phosphorylated RIPK3 and MLKL in primary neurons after OGD" to make it more descriptive, lines 236-237.

3. Background information of OTUB2 should be provided in the Introduction.

Response: As suggested by the reviewer, we have added the background information of OTUB2 in the Introduction section, lines 78-84.

4. MACO, OTUB2, OGD, etc., should be spelled out when they first appear in the Introduction and Results. The reason why HT22 cell was used should be explained when it first appears in the Results.

Response: We thank the reviewer for these constructive suggestions. We have spelled out the abbreviations when they first appear, lines 26 (OTUB2), 80 (OTUB2), 103-104 (MCAO), 135 (LPS), 154-155 (TNF- α , CHX), 170-171 (OGD). The HT22 cell line is widely used to study neuronal death (Feng et al Nat Commun. 2014; Karuppagounder et al Sci Transl Med. 2016). We have stated the reason why the HT22 cell line was used when it first appears, lines 151-153.

5. Reduction of K48 ubiquitination of RIPK3 by OTUB2 was shown in Figure 5. But the authors also need to show whether OTUB2 had effects on other types of ubiquitinations

on RIPK3.

Response: It has been shown that RIPK3 can be modified by K33, K48, and K63-specific polyubiquitin chains (Choi et al. *Mol Cell*. 2018; Lee et al. *Nat Cell Biol*. 2019; Zhou et al. *J Crohns Colitis*. 2021). In addition, K11-linked polyubiquitination can also induce protein degradation (Wang et al. *Cell Death Differ*. 2024). Therefore, we analyzed the influence of OTUB2 on K11, K33, K48, K63, and total ubiquitination of RIPK3. The results demonstrate that OTUB2 reduces K48 and total ubiquitination of RIPK3 but has no impact on K11, K33, and K63 ubiquitination of RIPK3 (Figure 5D-E, Appendix Figure S10A-C).

References

Feng X et al. Receptor-interacting protein 140 attenuates endoplasmic reticulum stress in neurons and protects against cell death. *Nat Commun*. 2014;5:4487.

Karuppagounder SS et al. Therapeutic targeting of oxygen-sensing prolyl hydroxylases abrogates ATF4-dependent neuronal death and improves outcomes after brain hemorrhage in several rodent models. *Sci Transl Med*. 2016;8(328):328ra29.

Choi SW et al. PELI1 Selectively Targets Kinase-Active RIP3 for Ubiquitylation-Dependent Proteasomal Degradation. *Mol Cell*. 2018;70(5):920-935.e7.

Lee SB et al. The AMPK-Parkin axis negatively regulates necroptosis and tumorigenesis by inhibiting the necrosome. *Nat Cell Biol*. 2019;21(8):940-951.

Zhou M et al. ABIN3 Negatively Regulates Necroptosis-induced Intestinal Inflammation Through Recruiting A20 and Restricting the Ubiquitination of RIPK3 in Inflammatory Bowel Disease. *J Crohns Colitis*. 2021;15(1):99-114.

Zhou M et al. ABIN3 Negatively Regulates Necroptosis-induced Intestinal Inflammation Through Recruiting A20 and Restricting the Ubiquitination of RIPK3 in Inflammatory Bowel Disease. *J Crohns Colitis*. 2021;15(1):99-114.

Wang Z et al. TRIM3 facilitates ferroptosis in non-small cell lung cancer through promoting SLC7A11/xCT K11-linked ubiquitination and degradation. *Cell Death Differ*. 2024;31(1):53-64.

13th Feb 2025

Dear Prof. Wang,

Thank you for the submission of your revised manuscript to EMBO Molecular Medicine. I am pleased to inform you that we will be able to accept your manuscript pending the following final amendments:

- 1) Authors: Please provide institutional email address for the corresponding author.
- 2) In the main manuscript file, please do the following:
 - Please address all comments suggested by our data editors listed below:
 - o Figure legends:
 1. Please note that the exact p values are not provided in the legends of figures 1B, F, J; 2F, 3B, I, K, O; 4B, 6A, E, F, G, H; 7C.
 2. Please note that the error bars are not defined in the legends of figures 1B, F, G-K. 2C, D, I-M; 3B-G, I, J-L, N, O; 4B, C; 6E-H; 7C, D, H, I.
 - Add callouts for Appendix Tables 2-17.
 - Move "Disclosure Statement & Competing Interests" after "Acknowledgement".
 - Please remove Reagents and Tools Table and uploaded it as a separate file. Structured Methods section includes Reagents and Tools Table followed by a Methods and Protocols section. More information on how to adhere to this format as well as downloadable templates (.docx) for the Reagents and Tools Table can be found in our author guidelines: <https://www.embopress.org/page/journal/17574684/authorguide#structuredmethods>
 - An example of a paper with Structured Methods can be found here: <https://www.embopress.org/doi/full/10.1038/s44320-024-00037-6#sec-4>
 - In Methods, provide the antibody dilutions that were used for each antibody.
- 3) Appendix: Please submit the file in PDF format.
- 4) Funding: Please make sure that information about all sources of funding are complete in both our submission system and in the manuscript. The Oujiang Laboratory (Zhejiang Lab for Regenerative Medicine, Vision, and Brain Health) and Wenzhou Medical University Scientific Research Center are missing in our submission system.
- 5) Please remove "Author contributions" file.
- 6) The Paper Explained: Please add it to the main manuscript file.
- 7) Synopsis:
 - Synopsis image: Please format the image to 550 px-wide x (300 - 600)-px high and upload it as a high-resolution JPEG file.
 - Please check your synopsis text and image before submission with your revised manuscript. Please be aware that in the proof stage minor corrections only are allowed (e.g., typos).
- 8) Source data: Please upload source data as one folder per figure for the main figures and a zip folder for all appendix figures and upload completed source data checklist.
- 9) As part of the EMBO Publications transparent editorial process initiative (see our Editorial at <http://embomolmed.embopress.org/content/2/9/329>), EMBO Molecular Medicine will publish online a Review Process File (RPF) to accompany accepted manuscripts. This file will be published in conjunction with your paper and will include the anonymous referee reports, your point-by-point response and all pertinent correspondence relating to the manuscript. Let us know whether you agree with the publication of the RPF and as here, if you want to remove or not any figures from it prior to publication. Please note that the Authors checklist will be published at the end of the RPF.
- 10) Please provide a point-by-point letter INCLUDING my comments as well as the reviewer's reports and your detailed responses (as Word file).

I look forward to reading a new revised version of your manuscript as soon as possible.

Yours sincerely,

Zeljko Durdevic

*** Instructions to submit your revised manuscript ***

*** PLEASE NOTE *** As part of the EMBO Publications transparent editorial process initiative (see our Editorial at

<https://www.embopress.org/doi/pdf/10.1002/emmm.201000094>), EMBO Molecular Medicine will publish online a Review Process File to accompany accepted manuscripts.

1) a .docx formatted version of the manuscript text (including Figure legends and tables)

2) Separate figure files*

3) supplemental information as Expanded View and/or Appendix. Please carefully check the authors guidelines for formatting Expanded view and Appendix figures and tables at <https://www.embopress.org/page/journal/17574684/authorguide#expandedview>

4) a letter INCLUDING the reviewer's reports and your detailed responses to their comments (as Word file).

5) The paper explained: EMBO Molecular Medicine articles are accompanied by a summary of the articles to emphasize the major findings in the paper and their medical implications for the non-specialist reader. Please provide a draft summary of your article highlighting

This may be edited to ensure that readers understand the significance and context of the research.

Please refer to any of our published articles for an example.

6) Author contributions: the contribution of every author must be detailed in a separate section.

7) EMBO Molecular Medicine now requires a complete author checklist (<https://www.embopress.org/page/journal/17574684/authorguide>) to be submitted with all revised manuscripts. Please use the checklist as guideline for the sort of information we need WITHIN the manuscript. The checklist should only be filled with page numbers where the information can be found. This is particularly important for animal reporting, antibody dilutions (missing) and exact values and n that should be indicated instead of a range.

8) Every published paper now includes a 'Synopsis' to further enhance discoverability. Synopses are displayed on the journal webpage and are freely accessible to all readers. They include a short stand first (maximum of 300 characters, including space) as well as 2-5 one sentence bullet points that summarise the paper. Please write the bullet points to summarise the key NEW findings. They should be designed to be complementary to the abstract - i.e. not repeat the same text. We encourage inclusion of key acronyms and quantitative information (maximum of 30 words / bullet point). Please use the passive voice. Please attach these in a separate file or send them by email, we will incorporate them accordingly.

You are also welcome to suggest a striking image or visual abstract to illustrate your article. If you do please provide a jpeg file 550 px-wide x 300-600px high.

9) A Conflict of Interest statement should be provided in the main text

10) Please note that we now mandate that all corresponding authors list an ORCID digital identifier. This takes <90 seconds to complete. We encourage all authors to supply an ORCID identifier, which will be linked to their name for unambiguous name identification.

Currently, our records indicate that the ORCID for your account is 0000-0001-8428-9339.

Link Not Available

11) Include a Reagents and Tools Table as part of the Methods section, which can be downloaded from our author guidelines

(<https://www.embopress.org/page/journal/17574684/authorguide#structuredmethods>)

Photos 400-800 DPI

*Additional important information regarding figures and illustrations can be found at

<https://bit.ly/EMBOPressFigurePreparationGuideline>. See also figure legend preparation guidelines:

<https://www.embopress.org/page/journal/17574684/authorguide#figureformat>

***** Reviewer's comments *****

Referee #1 (Remarks for Author):

All my concerns have been addressed.

Referee #3 (Remarks for Author):

The authors have adequately addressed my comments

The authors addressed the remaining editorial issues.

19th Feb 2025

Dear Prof. Wang,

We are pleased to inform you that your manuscript is accepted for publication and is now being sent to our publisher to be included in the next available issue of EMBO Molecular Medicine.

Zeljko Durdevic
Senior Editor
EMBO Molecular Medicine
